# STEIN LATENT OPTIMIZATION
# FOR GENERATIVE ADVERSARIAL NETWORKS

**Uiwon Hwang**[1], **Heeseung Kim**[1], **Dahuin Jung**[1], **Hyemi Jang**[1], **Hyungyu Lee**[1], **Sungroh Yoon**[1,2,*]
[1]Department of Electrical and Computer Engineering, [2]AIIS, ASRI, INMC, ISRC, NSI, and
Interdisciplinary Program in Artificial Intelligence, Seoul National University, Seoul 08826, Korea
`{uiwon.hwang, gmltmd789, annajung0625, wkdal9512, rucy74, sryoon}@snu.ac.kr`

## ABSTRACT

Generative adversarial networks (GANs) with clustered latent spaces can perform
conditional generation in a completely unsupervised manner. In the real world, the
salient attributes of unlabeled data can be imbalanced. However, most of existing
unsupervised conditional GANs cannot cluster attributes of these data in their latent
spaces properly because they assume uniform distributions of the attributes. To
address this problem, we theoretically derive Stein latent optimization that pro-
vides reparameterizable gradient estimations of the latent distribution parameters
assuming a Gaussian mixture prior in a continuous latent space. Structurally, we
introduce an encoder network and novel unsupervised conditional contrastive loss
to ensure that data generated from a single mixture component represent a single
attribute. We confirm that the proposed method, named Stein Latent Optimization
for GANs (SLOGAN), successfully learns balanced or imbalanced attributes and
achieves state-of-the-art unsupervised conditional generation performance even
in the absence of attribute information (e.g., the imbalance ratio). Moreover, we
demonstrate that the attributes to be learned can be manipulated using a small
amount of probe data.

## 1 INTRODUCTION

GANs have shown remarkable results in the synthesis of realistic data conditioned on a specific class
(Odena et al., 2017; Miyato & Koyama, 2018; Kang & Park, 2020). Training conditional GANs
requires a massive amount of labeled data; however, data are often unlabeled or possess only a few
labels. For unsupervised conditional generation, the salient attributes of the data are first identified by
unsupervised learning and used for conditional generation of data. Recently, several unsupervised
conditional GANs have been proposed (Chen et al., 2016; Mukherjee et al., 2019; Pan et al., 2021;
Armandpour et al., 2021). By maximizing a lower bound of mutual information between latent codes
and generated data, they cluster the attributes of the underlying data distribution in their latent spaces.
These GANs achieve satisfactory performance when the salient attributes of data are balanced.

However, the attributes of real-world data can be *imbalanced*. For example, in the CelebA dataset
(Liu et al., 2015), examples with one attribute (not wearing eyeglasses) outnumber the other attribute
(wearing eyeglasses). Similarly, the number of examples with disease-related attributes in a biomed-
ical dataset might be miniscule (Hwang et al., 2019). Thus, the imbalanced nature of real-world
attributes must be considered for unsupervised conditional generation. Most of existing unsupervised
conditional GANs are not suitable for real-world attributes, because they assume balanced attributes if
the imbalance ratio is unknown (Chen et al., 2016; Mukherjee et al., 2019; Pan et al., 2021). Examples
where existing methods fail to learn imbalanced attributes are shown in Figure 1 (a), (b) and (c).

In this paper, we propose unsupervised conditional GANs, referred to as Stein Latent Optimization
for GANs (SLOGAN). We define the latent distribution of the GAN models as Gaussian mixtures to
enable the imbalanced attributes to be naturally clustered in a continuous latent space. We derive
reparameterizable gradient identities for the mean vectors, full covariance matrices, and mixing
coefficients of the latent distribution using Stein's lemma. This enables stable learning and makes

---
*Corresponding author

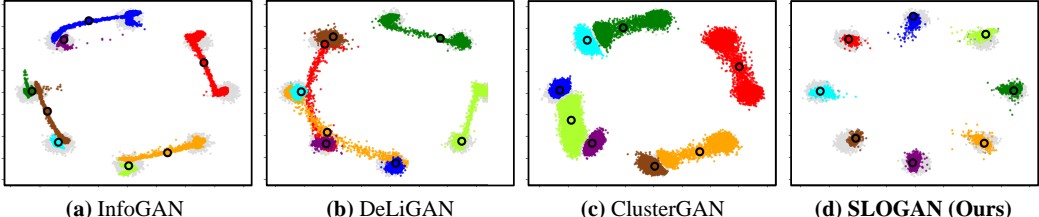

| **(a)** InfoGAN | **(b)** DeLiGAN | **(c)** ClusterGAN | **(d)** SLOGAN (Ours) |

Figure 1: Unsupervised conditional generation on synthetic dataset. Dataset consists of eight two-dimensional Gaussians (gray dots), and the number of unlabeled data instances from each Gaussian distribution is imbalanced (clockwise from the top, imbalance ratio between the first four Gaussians and the remaining four is 1:3). It is considered that the instances sampled from the same Gaussian share an attribute. Dots with different colors denote the data generated from different latent codes. Bold circles represent the samples generated from the mean vectors of latent distributions.

latent distribution parameters, including the mixing coefficient, learnable. We then devise a GAN framework with an encoder network and an unsupervised conditional contrastive loss (U2C loss), which can interact well with the learnable Gaussian mixture prior (Figure 2). This framework facilitates the association of data generated from a Gaussian component with a single attribute.

For the synthetic dataset, our method (Figure 1 (d)) shows superior performance on unsupervised conditional generation, with the accurately learned mixing coefficients. We performed experiments on various real-world datasets including MNIST (LeCun et al., 1998), Fashion-MNIST (Xiao et al., 2017), CIFAR-10 (Krizhevsky et al., 2009), CelebA (Liu et al., 2015), CelebA-HQ (Karras et al., 2017), and AFHQ (Choi et al., 2020) using architectures such as DCGAN (Radford et al., 2016), ResGAN (Gulrajani et al., 2017), and StyleGAN2 (Karras et al., 2020). Through experiments, we verified that the proposed method outperforms existing unsupervised conditional GANs in unsupervised conditional generation on datasets with balanced or imbalanced attributes. Furthermore, we confirmed that we could control the attributes to be learned when a small set of probe data is provided.

The contributions of this work are summarized as follows:

- We propose novel Stein Latent Optimization for GANs (SLOGAN). To the best of our knowledge, this is one of the first methods that can perform unsupervised conditional generation by considering the imbalanced attributes of real-world data.

- To enable this, we derive the implicit reparameterization for Gaussian mixture prior using Stein's lemma. Then, we devise a GAN framework with an encoder and an unsupervised conditional contrastive loss (U2C loss) suitable for implicit reparameterization.

- SLOGAN significantly outperforms the existing methods on unsupervised learning tasks, such as cluster assignment, unconditional data generation, and unsupervised conditional generation, on datasets that include balanced or imbalanced attributes.

## 2 BACKGROUND

### 2.1 GENERATIVE ADVERSARIAL NETWORKS

**Unsupervised conditional generation**  Several models including InfoGAN (Chen et al., 2016), ClusterGAN (Mukherjee et al., 2019), Self-conditioned GAN (Liu et al., 2020), CD-GAN (Pan et al., 2021), and PGMGAN (Armandpour et al., 2021) have been proposed to perform conditional generation in a completely unsupervised manner. However, these models primarily have two drawbacks: (1) Most of these methods embed the attributes in discrete variables, which induces discontinuity among the embedded attributes. (2) Most of them assume uniform distributions of the attributes, and thus fail to learn the imbalance in attributes when the imbalance ratio is not provided. In Appendix B.2, we discuss the above models in detail. By contrast, our work addresses the aforementioned limitations by combining GANs with the gradient estimation of the Gaussian mixture prior via Stein's lemma and representation learning on the latent space.

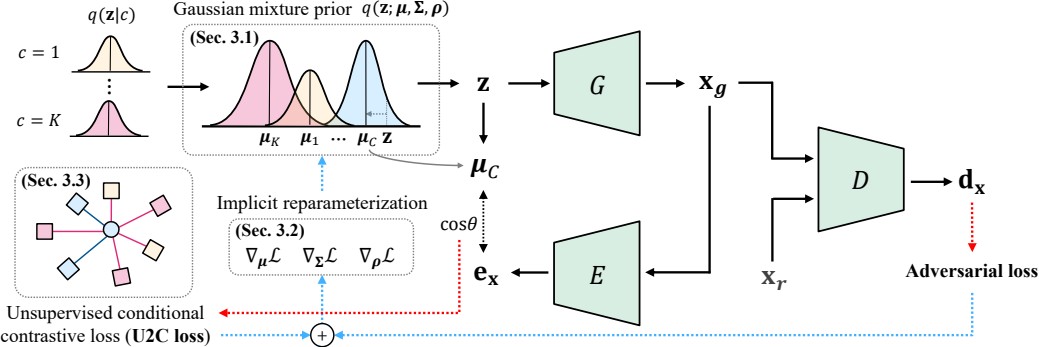

Figure 2: Overview of the SLOGAN model. Here, $\mathbf{x}_g$ denotes the data generated from a latent vector $\mathbf{z}$, $\mathbf{x}_r$ is a real data that is used for adversarial learning, and $C$ indicates a component ID of the Gaussian mixture prior with the highest responsibility $\text{argmax}_c\, q(c|\mathbf{z})$.

**GANs with Gaussian mixture prior** DeLiGAN (Gurumurthy et al., 2017) is analogous to the proposed method, as it assumes a Gaussian mixture prior and learns the mean vectors and covariance matrices via the reparameterization trick. However, DeLiGAN assumes uniform mixing coefficients without updating them. As a result, it fails to perform unsupervised conditional generation on datasets with imbalanced attributes. In addition, it uses the explicit reparameterization trick, which inevitably suffers from high variance in the estimated gradients. This will be discussed further in Section 2.3.

## 2.2 CONTRASTIVE LEARNING

Contrastive learning aims to learn representations by contrasting neighboring with non-neighboring instances (Hadsell et al., 2006). In general, contrastive loss is defined as a critic function that approximates the log density ratio $\log p(y|x)/p(y)$ of two random variables $X$ and $Y$. By minimizing the loss, the lower bound of the mutual information $I(X;Y)$ is approximately maximized (Poole et al., 2019). Several studies have shown that contrastive losses are advantageous for the representation learning of imbalanced data (Kang et al., 2021; 2020; Wanyan et al., 2021). Motivated by these observations, we propose a contrastive loss that cooperates with a learnable latent distribution.

## 2.3 GRADIENT ESTIMATION FOR GAUSSIAN MIXTURE

**Stein's lemma** Stein's lemma provides a first-order gradient identity for a multivariate Gaussian distribution. The univariate case of Stein's lemma can be described as follows:

**Lemma 1.** *Let function $h(\cdot) : \mathbb{R} \mapsto \mathbb{R}$ be continuously differentiable. $q(z)$ is a univariate Gaussian distribution parameterized by the mean $\mu$ and variance $\sigma$. Then, the following identity holds:*

$$\mathbb{E}_{q(z)}\left[\sigma^{-1}(z-\mu)h(z)\right] = \mathbb{E}_{q(z)}\left[\nabla_z h(z)\right] \tag{1}$$

Lin et al. (2019b) generalized Stein's lemma to exponential family mixtures and linked it to the implicit reparameterization trick. Stein's lemma has been applied to various fields of deep learning, including Bayesian deep learning (Lin et al., 2019a) and adversarial robustness (Wang et al., 2020). To the best of our knowledge, our work is the first to apply Stein's lemma to GANs.

**Reparameterization trick** A simple method to estimate gradients of the parameters of Gaussian mixtures is explicit reparameterization, used in DeLiGAN. When the $c$-th component is selected according to the mixing coefficient $p(c)$, the latent variable is calculated as follows: $\mathbf{z} = \boldsymbol{\mu}_c + \boldsymbol{\epsilon} \cdot \boldsymbol{\Sigma}_c^{1/2}$, where $\boldsymbol{\epsilon} \sim \mathcal{N}(\mathbf{0}, \mathbf{I})$. Derivatives of a loss function $\frac{\partial \mathcal{L}(z)}{\partial \boldsymbol{\mu}}$ and $\frac{\partial \mathcal{L}(z)}{\partial \boldsymbol{\Sigma}}$ only update the mean and covariance matrices of the *selected* ($c$-th) component $\boldsymbol{\mu}_c$ and $\boldsymbol{\Sigma}_c$, respectively. Gradient estimation using explicit reparameterization is unbiased; however, it has a distinctly high variance. For a single latent vector $\mathbf{z}$, the implicit reparameterization trick (Figurnov et al., 2018) updates the parameters of *all* the latent components. Gradient estimation using implicit reparameterization is unbiased and has a lower variance, which enables a more stable and faster convergence of the model. The gradients for the parameters of the Gaussian mixture prior in our method are implicitly reparameterizable.

## 3 PROPOSED METHOD

In the following sections, we propose Stein Latent Optimization for GANs (SLOGAN). We assume a Gaussian mixture prior (Section 3.1), derive implicit reparameterization of the parameters of the mixture prior (Section 3.2), and construct a GAN framework with U2C loss (Section 3.3). Additionally, we devise a method to manipulate attributes to be learned if necessary (Section 3.4). An overview of SLOGAN is shown in Figure 2.

### 3.1 GAUSSIAN MIXTURE PRIOR

We consider a GAN with a generator $G : \mathbb{R}^{d_z} \mapsto \mathbb{R}^{d_x}$ and a discriminator $D : \mathbb{R}^{d_x} \mapsto \mathbb{R}$, where $d_z$ and $d_x$ are the dimensions of latent and data spaces, respectively. In the latent space $\mathcal{Z} \in \mathbb{R}^{d_z}$, we consider a conditional latent distribution $q(\mathbf{z}|c) = \mathcal{N}(\mathbf{z}; \boldsymbol{\mu}_c, \Sigma_c)$, $c = 1, ..., K$, where $K$ is the number of components we initially set and $\boldsymbol{\mu}_c, \Sigma_c$ are the mean vector and covariance matrix of the $c$-th component, respectively. Subsequently, we consider a Gaussian mixture $q(\mathbf{z}) = \sum_{c=1}^{K} p(c)q(\mathbf{z}|c)$ parameterized by $\boldsymbol{\mu} = \{\boldsymbol{\mu}_c\}_{c=1}^{K}$, $\boldsymbol{\Sigma} = \{\Sigma_c\}_{c=1}^{K}$ and $\boldsymbol{\pi} = \{\pi_c\}_{c=1}^{K} = \{p(c)\}_{c=1}^{K}$ as the prior.

We hypothesize that a mixture prior in a continuous space could model some continuous attributes of real-world data (e.g., hair color) more naturally than categorical priors which could introduce discontinuity (Mukherjee et al., 2019). Because we use implicit reparameterization of a mixture of Gaussian priors (derived in Section 3.2), SLOGAN can fully benefit from implicit reparameterization and U2C loss. By contrast, the implicit reparameterization of prior distributions that do not belong to the exponential family (e.g., categorical priors) remains an open question.

In the experiments, the elements of $\boldsymbol{\mu}_c$ were sampled from $\mathcal{N}(0, 0.1)$, and we selected $\Sigma_c = I$ and $\pi_c = 1/K$ as the initial values. For the convenience of notation, we define the latent distribution $q = q(\mathbf{z})$, the mixing coefficient $\pi_c = p(c)$, and $\boldsymbol{\delta}(\mathbf{z}) = \{\delta(\mathbf{z})_c\}_{c=1}^{K}$, where $\delta(\mathbf{z})_c = q(\mathbf{z}|c)/q(\mathbf{z})$. $q(c|\mathbf{z})$, the responsibility of component $c$ for a latent vector $\mathbf{z}$, can be expressed as follows:

$$q(c|\mathbf{z}) = \frac{q(c, \mathbf{z})}{q(\mathbf{z})} = \frac{q(\mathbf{z}|c)p(c)}{q(\mathbf{z})} = \delta(\mathbf{z})_c \pi_c \qquad (2)$$

### 3.2 GRADIENT IDENTITIES

We present gradient identities for the latent distribution parameters. To derive the identities, we use the generalized Stein's lemma for Gaussian mixtures with full covariance matrices (Lin et al., 2019b). First, we derive a gradient identity for the mean vector using Bonnet's theorem (Bonnet, 1964).

**Theorem 1.** *Given an expected loss of the generator $\mathcal{L}$ and a loss function for a sample $\ell(\cdot) : \mathbb{R}^{d_z} \mapsto \mathbb{R}$, we assume $\ell$ to be continuously differentiable. Then, the following identity holds:*

$$\nabla_{\boldsymbol{\mu}_c} \mathcal{L} = \mathbb{E}_q \left[ \delta(\mathbf{z})_c \pi_c \nabla_{\mathbf{z}} \ell(\mathbf{z}) \right] \qquad (3)$$

Proof of Theorem 1 is given in Appendix C.1.

We derive a gradient identity for the covariance matrix via Price's theorem (Price, 1958). Among the two versions of the Price's theorem, we use the first-order identity to minimize computational cost.

**Theorem 2.** *With the same assumptions as in Theorem 1, the following gradient identity holds:*

$$\nabla_{\Sigma_c} \mathcal{L} = \frac{1}{2} \mathbb{E}_q \left[ \delta(\mathbf{z})_c \pi_c \Sigma_c^{-1} \left( \mathbf{z} - \boldsymbol{\mu}_c \right) \nabla_{\mathbf{z}}^T \ell(\mathbf{z}) \right] \qquad (4)$$

Proof of Theorem 2 is given in Appendix C.2. In the implementation, we replaced the expectation of the right-hand side of Equation 4 with the average for a batch of latent vectors; hence, the updated $\Sigma_c$ may not be symmetric or positive-definite. To force a valid covariance matrix, we modify the updates of the covariance matrix as follows:

$$\Delta\Sigma_c = -\nabla_{\Sigma_c} \mathcal{L} = -\frac{1}{2} \mathbb{E}_q \left[ \frac{1}{2} \left( S_{\mathbf{z}} + S_{\mathbf{z}}^T \right) \right] \qquad (5)$$

$$\Delta\Sigma_c' = \Delta\Sigma_c + \frac{\gamma}{2} \Delta\Sigma_c \Sigma_c^{-1} \Delta\Sigma_c \qquad (6)$$

**Algorithm 1** Training procedure of SLOGAN

Initialize $\boldsymbol{\mu}$, $\boldsymbol{\Sigma}$, $\boldsymbol{\rho}$, parameters of $D$, $G$, and $E$
**while** training loss is not converged **do**
    Sample a batch of data $\{\mathbf{x}^i\}_{i=1}^B \sim p(\mathbf{x})$
    Sample a batch of latent vectors $\{\mathbf{z}^i\}_{i=1}^B \sim q(\mathbf{z})$
    **for** $i = 1, ..., B$ **do**
        Calculate $\ell_{\text{adv}}(\mathbf{z}^i)$ and $\ell_{\text{U2C}}(\mathbf{z}^i)$ for a latent vector $\mathbf{z}^i$
        $S_{\mathbf{z}^i} \leftarrow \delta(\mathbf{z}^i)_c \pi_c \Sigma_c^{-1}(\mathbf{z}^i - \boldsymbol{\mu}_c)\nabla_{\mathbf{z}^i}^T(\ell_{\text{adv}}(\mathbf{z}^i) + \lambda\ell_{\text{U2C}}(\mathbf{z}^i))$
    **end for**
    **for** $c = 1, ..., K$ **do**
        Update $\boldsymbol{\mu}_c$, $\boldsymbol{\Sigma}_c$ and $\boldsymbol{\rho}_c$ via stochastic gradient estimation
        $\boldsymbol{\mu}_c \leftarrow \boldsymbol{\mu}_c - \gamma\frac{1}{B}\sum_{i=1}^B \delta(\mathbf{z}^i)_c \pi_c \nabla_{\mathbf{z}^i}(\ell_{\text{adv}}(\mathbf{z}^i) + \lambda\ell_{\text{U2C}}(\mathbf{z}^i))$
        $\Delta\Sigma_c \leftarrow -\frac{1}{4B}\sum_{i=1}^B \left(S_{\mathbf{z}^i} + S_{\mathbf{z}^i}^T\right)$
        $\Sigma_c \leftarrow \Sigma_c + \gamma\left(\Delta\Sigma_c + \frac{\gamma}{2}\Delta\Sigma_c\Sigma_c^{-1}\Delta\Sigma_c\right)$
        $\rho_c \leftarrow \rho_c - \gamma\frac{1}{B}\sum_{i=1}^B \pi_c\left(\delta(\mathbf{z}^i)_c - 1\right)\ell_{\text{adv}}(\mathbf{z}^i)$
    **end for**
    Update $G$, $E$ and $D$ using SGD
    $\nabla_{G,E}\frac{1}{B}\sum_{i=1}^B \left(\ell_{\text{adv}}(\mathbf{z}^i) + \lambda\ell_{\text{U2C}}(\mathbf{z}^i)\right)$
    $\nabla_D\left(-\frac{1}{B}\sum_{i=1}^B \ell_{\text{adv}}(\mathbf{z}^i) - \frac{1}{B}\sum_{i=1}^B D(\mathbf{x}^i)\right)$
**end while**

**Algorithm 2** Intra-cluster FID

**input** : $\{\{\mathbf{x}_y^i\}_{i=1}^N\}_{y=1}^K$ - Data sampled from $p(\mathbf{x}|y)$ for $y = 1, ..., K$;
      $\{\{\mathbf{z}_c^i\}_{i=1}^N\}_{c=1}^K$ - Latent vectors sampled from $q(\mathbf{z}|c)$ for $c = 1, ..., K$
**output** : ICFID - Intra-cluster FID;
      $Y_c$ - Class-cluster assignments
$Y \leftarrow \{1, ..., K\}$
$C \leftarrow \{1, ..., K\}$
**for** each class $y$ in $Y$ **do**
    $\mathbf{X}_r \leftarrow \{\mathbf{x}_y^i\}_{i=1}^N$
    **for** each cluster $c$ in $C$ **do**
        $\mathbf{X}_g \leftarrow \{\mathbf{x}_c^i\}_{i=1}^N$
        $d(y, c) \leftarrow \text{FID}(\mathbf{X}_r, \mathbf{X}_g)$
    **end for**
    $c^* \leftarrow \text{argmin}_{c\in C} d(y, c)$
    $\text{ICFID}(y) \leftarrow d(y, c^*)$
    $Y_c(y) \leftarrow c^*$
    Remove $c^*$ from $C$
**end for**
$\text{ICFID} \leftarrow \frac{1}{K}\sum_{y=1}^K \text{ICFID}(y)$

where $S_{\mathbf{z}} = \delta(\mathbf{z})_c \pi_c \Sigma_c^{-1}(\mathbf{z} - \boldsymbol{\mu}_c)\nabla_{\mathbf{z}}^T\ell(\mathbf{z})$, and $\gamma$ denotes the learning rate for $\Sigma_c$. Equation 5 holds as $\Delta\Sigma_c = \frac{1}{2}E_q[S_{\mathbf{z}}] = \frac{1}{2}E_q[S_{\mathbf{z}}^T]$. Motivated by Lin et al. (2020), Equation 6 ensures the positive-definiteness of the covariance matrix, which is proved by Theorem 3.

**Theorem 3.** *The updated covariance matrix* $\Sigma_c' = \Sigma_c + \gamma\Delta\Sigma_c'$ *with the modified update rule specified in Equation 6 is positive-definite if* $\Sigma_c$ *is positive-definite.*

Proof of Theorem 3 is provided in Appendix C.3.

We introduce a mixing coefficient parameter $\rho_c$, which is updated instead of the mixing coefficient $\pi_c$, to guarantee that the updated mixing coefficients are non-negative and summed to one. $\pi_c$ can be calculated using the softmax function (i.e., $\pi_c = \exp(\rho_c)/\sum_{i=1}^K \exp(\rho_i)$). We can then derive the gradient identity for the mixing coefficient parameter as follows:

**Theorem 4.** *Let* $\rho_c$ *be a mixing coefficient parameter. Then, the following gradient identity holds:*

$$\nabla_{\rho_c}\mathcal{L} = \mathbb{E}_q\left[\pi_c\left(\delta(\mathbf{z})_c - 1\right)\ell(\mathbf{z})\right] \tag{7}$$

Proof of Theorem 4 is given in Appendix C.4. Because the gradients of the latent vector with respect to the latent parameters are computed by implicit differentiation via Stein's lemma, we obtain the implicit reparameterization gradients introduced by Figurnov et al. (2018).

### 3.3 CONTRASTIVE LEARNING

We introduce new unsupervised conditional contrastive loss (U2C loss) to learn salient attributes from data and to facilitate unsupervised conditional generation. We consider a batch of latent vectors $\{\mathbf{z}^i\}_{i=1}^B \sim q(\mathbf{z})$, where $B$ is the batch size. Generator $G$ receives the $i$-th latent vector $\mathbf{z}^i$ and generates data $\mathbf{x}_g^i = G(\mathbf{z}^i)$. The adversarial loss for $G$ with respect to the sample $\mathbf{z}^i$ is as follows:

$$\ell_{\text{adv}}(\mathbf{z}^i) = -D(G(\mathbf{z}^i)) \tag{8}$$

We also introduce an encoder network $E$ to implement U2C loss. The synthesized data $\mathbf{x}_g^i$ enters $E$, and $E$ generates an encoded vector $\mathbf{e}_{\mathbf{x}}^i = E(\mathbf{x}_g^i)$. Then, we find the mean vector $\boldsymbol{\mu}_C^i$, where $C$ is the component ID with the highest responsibility $q(c|\mathbf{z}^i)$. We calculate $C$ first because a generated sample should have the attribute of the most responsible component among multiple components in the continuous space. Second, to update the parameters of the prior using implicit reparameterization, the loss should be a function of a latent vector $\mathbf{z}^i$, as proved in Theorems 1, 2, and 4. The component ID for each sample is calculated as follows:

$$C^i = \underset{c}{\text{argmax}}\, q(c|\mathbf{z}^i) = \underset{c}{\text{argmax}}\, \delta(\mathbf{z}^i)_c \pi_c \tag{9}$$

where $q(c|\mathbf{z}^i) = \delta(\mathbf{z}^i)_c \pi_c$ is derived from Equation 2. To satisfy the assumption of the continuously differentiable loss function in Theorems 1 and 2, we adopt the Gumbel-Softmax relaxation (Jang et al., 2017), instead of the $\mathrm{argmax}$ function. We use $\boldsymbol{\mu}_{\mathbf{C}}^i = \sum_{c=1}^K \mathbf{C}_c^i \boldsymbol{\mu}_c$ to calculate U2C loss to ensure that the loss function is continuously differentiable with respect to $\mathbf{z}^i$, where $\mathbf{C}^i = \mathrm{Gumbel\text{-}Softmax}_\tau(\boldsymbol{\delta}(\mathbf{z}^i)\boldsymbol{\pi})$ and $\tau = 0.01$. We derive U2C loss as follows:

$$\ell_{\mathrm{U2C}}(\mathbf{z}^i) = -\log \frac{\exp(\cos\theta_{ii})}{\frac{1}{B}\sum_{j=1}^B \exp(\cos\theta_{ij})} \tag{10}$$

where we select the cosine similarity between $\mathbf{e}_{\mathbf{x}}^i$ and $\boldsymbol{\mu}_{\mathbf{C}}^j$, $\cos\theta_{ij} = \mathbf{e}_{\mathbf{x}}^i \cdot \boldsymbol{\mu}_{\mathbf{C}}^j / \|\mathbf{e}_{\mathbf{x}}^i\| \|\boldsymbol{\mu}_{\mathbf{C}}^j\|$ as the critic function that approximates the log density ratio $\log p(C^j|\mathbf{x}_g^i)/p(C^j)$ for contrastive learning. Given a test data, the probability for each cluster can be calculated using the assumption of the critic function, which enables us to assign a cluster for the data. Cluster assignment is described in Appendix D.2.

Intuitively, a mean vector $\boldsymbol{\mu}_{\mathbf{C}}^i$ of a latent mixture component is regarded as a prototype of each attribute. U2C loss encourages the encoded vector $\mathbf{e}_{\mathbf{x}}^i$ of the generated sample to be similar to its assigned low-dimensional prototype $\boldsymbol{\mu}_{\mathbf{C}}^i$ in the latent space. This allows each salient attribute clusters in the latent space, and each component of the learned latent distribution is responsible for a certain attribute of the data. If $\cos\theta_{ii}$ is proportional to the log density ratio $\log p(C^i|\mathbf{x}_g^i)/p(C^i)$, minimizing U2C loss in Equation 10 is equivalent to maximizing the lower bound of the mutual information $I(C^i; \mathbf{x}_g^i)$, as discussed by Poole et al. (2019) and Zhong et al. (2020).

$G$ and $E$ are trained to minimize $\frac{1}{B}\sum_{i=1}^B \left(\ell_{\mathrm{adv}}(\mathbf{z}^i) + \lambda\ell_{\mathrm{U2C}}(\mathbf{z}^i)\right)$, where $\lambda$ denotes the coefficient of U2C loss. Both $\boldsymbol{\mu}$ and $\boldsymbol{\Sigma}$ are learned by substituting $\ell_{\mathrm{adv}}(\mathbf{z}^i) + \lambda\ell_{\mathrm{U2C}}(\mathbf{z}^i)$ into $\ell$ of Equations 3 and 6, respectively. When U2C loss is used to update $\boldsymbol{\pi}$, U2C loss hinders $\boldsymbol{\pi}$ from estimating the imbalance ratio of attributes in the data well, which is discussed in Appendix A.3 with a detailed explanation and an empirical result. Therefore, $\boldsymbol{\rho}$, from which $\boldsymbol{\pi}$ is calculated, uses only the adversarial loss, and $\ell$ of Equation 7 is substituted by $\ell_{\mathrm{adv}}(\mathbf{z}^i)$. $\boldsymbol{\mu}$, $\boldsymbol{\Sigma}$ and $\boldsymbol{\rho}$ are learned using a batch average of estimated gradients, which is referred to as stochastic gradient estimation, instead of expectation over the latent distribution $q$. The entire training procedure of SLOGAN is presented in Algorithm 1.

To help that the latent space does not learn low-level attributes, such as background color, we additionally used the SimCLR (Chen et al., 2020) loss on the generated data with DiffAugment (Zhao et al., 2020) to train the encoder on colored image datasets. Methodological details and discussion on SimCLR are presented in Appendix D.4 and A.3, respectively.

### 3.4 ATTRIBUTE MANIPULATION

For datasets such as face attributes, a data point can have multiple attributes simultaneously. To learn a desired attribute from such data, a probe dataset $\{\mathbf{x}_c^i\}_{i=1}^M$ for the $c$-th latent component, which consists of $M$ data points with the desired attribute, can be utilized. We propose the following loss:

$$\mathcal{L}_{\mathrm{m}} = \frac{1}{M}\sum_{i=1}^M -\log \frac{\exp(\cos\theta_c^i)}{\sum_{k=1}^K \exp(\cos\theta_k^i)} \tag{11}$$

where $\cos\theta_k^i = E(\mathbf{x}_c^i) \cdot \boldsymbol{\mu}_k / \|E(\mathbf{x}_c^i)\| \|\boldsymbol{\mu}_k\|$ is the cosine similarity between $E(\mathbf{x}_c^i)$ and $\boldsymbol{\mu}_k$. Our model manipulates attributes by minimizing $\mathcal{L}_{\mathrm{m}}$ for $\boldsymbol{\mu}$, $\boldsymbol{\Sigma}$, $G$, and $E$. In addition, mixup (Zhang et al., 2018) can be used to better learn attributes from a small probe dataset. The advantage of SLOGAN in attribute manipulation is that it can learn imbalanced attributes even if the attributes in the probe dataset are balanced, and perform better conditional generation. The detailed procedure of attribute manipulation is described in Appendix D.3.

## 4 EXPERIMENTS

### 4.1 DATASETS

We used the MNIST (LeCun et al., 1998), Fashion-MNIST (FMNIST) (Xiao et al., 2017), CIFAR-10 (Krizhevsky et al., 2009), CelebA (Liu et al., 2015), CelebA-HQ (Karras et al., 2017), and AFHQ (Choi et al., 2020) datasets to evaluate the proposed method. We also constructed some datasets with

Table 1: Performance comparison on balanced attributes

| Dataset | Metric | InfoGAN | DeLiGAN | DeLiGAN+ | ClusterGAN | SCGAN | CD-GAN | PGMGAN | **SLOGAN** |
|---|---|---|---|---|---|---|---|---|---|
| FMNIST | NMI ↑ | 0.64±0.02 | 0.64±0.03 | 0.57±0.07 | 0.61±0.03 | 0.56±0.01 | 0.56±0.04 | 0.47±0.01 | **0.66±0.01** |
| | FID ↓ | 5.28±0.12 | 6.65±0.48 | 7.23±0.56 | 6.32±0.25 | **5.07±0.19** | 9.05±0.11 | 9.13±0.28 | 5.20±0.36 |
| | ICFID ↓ | 32.18±2.11 | 34.87±5.29 | 30.53±8.71 | 37.20±5.50 | 26.23±7.10 | 36.61±0.47 | 40.00±4.38 | **23.31±2.77** |
| CIFAR-10 | NMI ↑ | 0.03±0.00 | 0.06±0.00 | 0.09±0.04 | 0.10±0.00 | 0.01±0.00 | 0.03±0.01 | 0.29±0.02 | **0.34±0.01** |
| | FID ↓ | 81.84±2.27 | 212.20±4.52 | 110.51±7.70 | 61.97±3.69 | 199.28±57.16 | 34.13±1.13 | 31.50±0.73 | **20.61±0.40** |
| | ICFID ↓ | 139.20±2.09 | 305.32±5.05 | 215.63±11.16 | 124.27±5.95 | 262.54±59.29 | 95.43±3.58 | 81.25±11.55 | **71.23±6.76** |

Table 2: Performance comparison on imbalanced attributes

| Dataset | Metric | InfoGAN | DeLiGAN | DeLiGAN+ | ClusterGAN | SCGAN | CD-GAN | PGMGAN | **SLOGAN** |
|---|---|---|---|---|---|---|---|---|---|
| FMNIST-5 | NMI ↑ | 0.58±0.07 | **0.68±0.05** | 0.65±0.01 | 0.60±0.02 | 0.60±0.06 | 0.59±0.01 | 0.24±0.02 | 0.66±0.06 |
| | FID ↓ | 5.40±0.14 | 7.05±0.49 | 6.33±0.44 | 5.61±0.17 | 9.34±0.56 | 11.80±0.43 | 5.29±0.16 | |
| | ICFID ↓ | 43.69±10.84 | 36.21±3.07 | 35.41±0.79 | 36.94±5.81 | 44.48±21.62 | 39.31±1.18 | 77.30±8.60 | **32.46±3.18** |
| CIFAR-2 (7:3) | NMI ↑ | 0.05±0.01 | 0.00±0.00 | 0.03±0.03 | 0.22±0.02 | 0.00±0.00 | 0.22±0.03 | 0.42±0.03 | **0.69±0.02** |
| | FID ↓ | 51.30±2.53 | 131.73±50.98 | 115.19±17.95 | 36.62±2.16 | 45.28±1.81 | 36.40±1.01 | 29.76±1.65 | **29.09±0.73** |
| | ICFID ↓ | 88.49±6.85 | 186.31±28.31 | 173.81±18.29 | 75.52±4.82 | 88.58±4.57 | 76.91±1.07 | 57.06±3.31 | **45.83±3.03** |

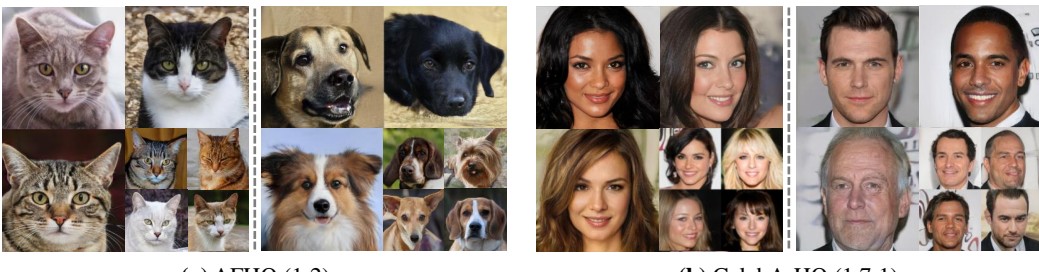

**(a)** AFHQ (1:2)    **(b)** CelebA-HQ (1.7:1)

Figure 3: Generated high-fidelity images from SLOGAN on (a) AFHQ and (b) CelebA-HQ.

imbalanced attributes. For example, we used two classes of the MNIST dataset (0 vs. 4, referred to as MNIST-2), two classes of the CIFAR-10 dataset (frogs vs. planes, referred to as CIFAR-2), and five clusters of the FMNIST dataset ({Trouser}, {Bag}, {T-shirt/top, Dress}, {Pullover, Coat, Shirt}, {Sneaker, Sandal, Ankle Boot}, referred to as FMNIST-5 with an imbalance ratio of 1:1:2:3:3). Details of the datasets are provided in Appendix E.

Although SLOGAN and other methods do not utilize labels for training, the data in experimental settings have labels predefined by humans. We consider that each class of dataset contains a distinct attribute. Thus, the model performance was measured using classes of datasets. The number of latent components or the dimension of the discrete latent code ($K$) was set as the number of classes of data.

## 4.2 EVALUATION METRICS

The performance of our method was evaluated quantitatively in three aspects: (1) whether the model could learn distinct attributes and cluster real data (i.e., cluster assignment), which is evaluated using normalized mutual information (NMI) (Mukherjee et al., 2019), (2) whether the overall data distribution $p(\mathbf{x}_r)$ could be estimated (i.e., unconditional data generation), which is measured using the Fréchet inception distance (FID) (Heusel et al., 2017), and, most importantly, (3) whether the data distribution for each attribute $p(\mathbf{x}_r|c)$ could be estimated (i.e., unsupervised conditional generation).

For unsupervised conditional generation, it is important to account for intra-cluster diversity as well as the quality of the generated samples. We introduce a modified version of FID named intra-cluster Fréchet inception distance (ICFID) described in Algorithm 2. We calculate FIDs between the real data of each class and generated data from each latent code (a mixture component for DeLiGAN and SLOGAN, and a category for other methods). We then greedily match a latent code with a class of real data with the smallest FID. We define ICFID as the average FID between the matched pairs and use it as an evaluation metric for unsupervised conditional generation. ICFID additionally provides class-cluster assignment (i.e., which cluster is the closest to the class).

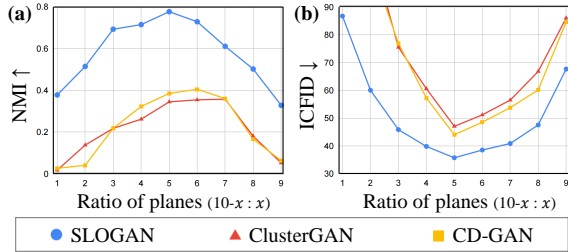

Figure 4: Performance comparison with respect to the imbalance ratio on (a) cluster assignment and (b) unsupervised conditional generation.

Table 3: Effectiveness of U2C loss

| Dataset | Ablation | ICFID $\downarrow$ |
|---|---|---|
| CIFAR-10 | SLOGAN w/o $\ell_{U2C}$ | 78.26 |
| | SLOGAN | **71.23** |
| MNIST-2 (7:3) | SLOGAN w/o $\ell_{U2C}$ | 9.43 |
| | SLOGAN | **5.91** |
| Synthetic | SLOGAN w/o $\ell_{U2C}$ | ✗ |
| | SLOGAN | ✓ |

Table 4: Effectiveness of implicit reparameterization

| Dataset | Ablation | $\pi_{y=0}$ (ground-truth: 0.7) | ICFID $\downarrow$ |
|---|---|---|---|
| | DeLiGAN with $\ell_{U2C}$ | 0.50 | 60.51 |
| CIFAR-2 (7:3) | DeLiGAN with $\ell_{U2C}$ and implicit $\rho$ update | 1.00 | 86.48 |
| | SLOGAN | **0.69** | **45.83** |

## 4.3 EVALUATION RESULTS

We compared SLOGAN with InfoGAN (Chen et al., 2016), DeLiGAN (Gurumurthy et al., 2017), ClusterGAN (Mukherjee et al., 2019), Self-conditioned GAN (SCGAN) (Liu et al., 2020), CD-GAN (Pan et al., 2021), and PGMGAN (Armandpour et al., 2021). Following Mukherjee et al. (2019), we used k-means clustering on the encoder outputs of the test data to calculate NMI. DeLiGAN has no encoder network; hence the pre-activation of the penultimate layer of $D$ was used for the clustering metrics. For a fair comparison, we also compared DeLiGAN with an encoder network (referred to as DeLiGAN+). The same network architecture and hyperparameters (e.g., learning rate) were used across all methods for comparison. Details of the experiments and DeLiGAN+ are presented in Appendices E and D.5, respectively.

**Balanced attributes** We compare SLOGAN with existing unsupervised conditional GANs on datasets with balanced attributes. As shown in Table 1 (The complete version is given in Appendix A.1.), SLOGAN outperformed other GANs, and comparisons with methods with categorical priors (ClusterGAN and CD-GAN) verified the advantages of the mixture priors.

**Imbalanced attributes** In Table 2 (The complete version is presented in Appendix A.2), we compare SLOGAN with existing methods on datasets with imbalanced attributes. ICFIDs of our method are much better than those of other methods, which indicates that SLOGAN was able to robustly capture the minority attributes in datasets and can generate data conditioned on the learned attributes. In CIFAR-2 (7:3), the ratio of frog and plane is 7 to 3 and the estimated $\pi$ is (0.69±0.02, 0.31±0.02), which are very close to the ground-truth (0.7, 0.3). Figure 3 (a) shows the images generated from each latent component of SLOGAN on AFHQ (Cat:Dog=1:2). More qualitative results are presented in Appendix A.7.

**Performance with respect to imbalance ratio** We compared the performance of SLOGAN with competitive benchmarks (ClusterGAN and CD-GAN) by changing the imbalance ratios of CIFAR-2 from 9:1 to 1:9. SLOGAN showed higher performance than the benchmarks on cluster assignment (Figure 4 (a)) and unsupervised conditional generation (Figure 4 (b)) for all imbalance ratios. Furthermore, our method shows a larger gap in ICFID with the benchmarks when the ratio of planes is low. This implies that SLOGAN works robustly in situations in which the attributes of data are highly imbalanced. We conducted additional experiments including interpolation in the latent space, benefits of ICFID. The results of the additional experiments are shown in Appendix A.

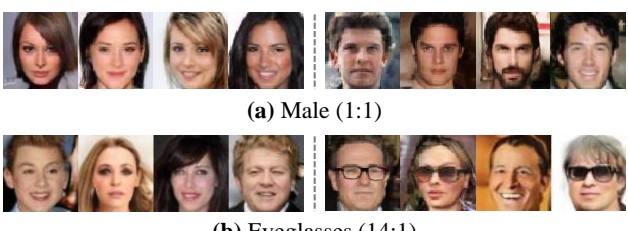

**(a)** Male (1:1)

**(b)** Eyeglasses (14:1)

Figure 5: Qualitative results of SLOGAN on CelebA.

Table 5: Quantitative results of SLOGAN on CelebA

| Imb. ratio | Male (1:1) | Eyeglasses (14:1) |
|---|---|---|
| NMI $\uparrow$ | 0.65±0.01 | 0.29±0.07 |
| FID $\downarrow$ | 5.18±0.20 | 5.83±0.44 |
| ICFID $\downarrow$ | 11.00±0.66 | 35.57±5.10 |
| $\pi_{y=0}$ | 0.56±0.02 | 0.82±0.04 |

### 4.4 ABLATION STUDY

**U2C loss** Table 3 (The complete version is given in Appendix A.3) shows the benefit of U2C loss on several datasets. Low-level features (e.g., color) of the CIFAR dataset differ depending on the class, which enables SLOGAN to function to some extent without U2C loss on CIFAR-10. In the MNIST dataset, the colors of the background (black) and object (white) are the same, and only the shape of objects differs depending on the class. U2C loss played an essential role on MNIST (7:3). The modes of the Synthetic dataset (Figure 1) are placed adjacent to each other, and SLOGAN cannot function on this dataset without U2C loss. From the results, we observed that the effectiveness of U2C loss depends on the properties of the datasets.

**Implicit reparameterization** To show the advantage of implicit over explicit reparameterization, we implemented DeLiGAN with U2C loss by applying explicit reparameterization on $\mu$ and $\Sigma$. Because the mixing coefficient cannot be updated with explicit reparameterization to the best of our knowledge, we also implemented DeLiGAN with U2C loss and implicit reparameterization on $\rho$ using Equation 7. In Table 4, SLOGAN using implicit reparameterization outperformed explicit reparameterization. When implicit $\rho$ update was added, the prior collapsed into a single component ($\pi_{y=0} = 1$) and ICFID increased. The lower variance of implicit reparameterized gradients prevents the prior from collapsing into a single component and improves the performance. Additional ablation studies and discussions are presented in Appendix A.3.

### 4.5 EFFECTS OF PROBE DATA

**CelebA + ResGAN** We demonstrate that SLOGAN can learn the desired attributes using a small amount of probe data. Among multiple attributes which co-exist in the CelebA dataset, we chose Male (1:1) and Eyeglasses (14:1). We randomly selected 30 probe images for each latent component. $\pi_{y=0}$ represents the learned mixing coefficient that correspond to the latent component associated with faces without the attribute. As shown in Figure 5 and Table 5, we observed that SLOGAN learned the desired attributes. Additional experiments on attribute manipulation are shown in Appendix A.3.

**CelebA-HQ + StyleGAN2** StyleGAN2 (Karras et al., 2020) differs from other GANs in that the latent vectors are used for *style*. Despite this difference, the implicit reparameterization and U2C loss can be applied to the input space of the mapping network. On the CelebA-HQ dataset, we used 30 male and 30 female faces as probe data. As shown in Figure 3 (b), SLOGAN successfully performed on high-resolution images and a recent architecture, even simultaneously with imbalanced attributes.

## 5 CONCLUSION

We have proposed a method called SLOGAN to generate data conditioned on learned attributes on real-world datasets with balanced or imbalanced attributes. We derive implicit reparameterization for the parameters of the latent distribution. We then proposed a GAN framework and unsupervised conditional contrastive loss (U2C loss). We verified that SLOGAN achieved state-of-the-art unsupervised conditional generation performance. In addition, a small amount of probe data helps SLOGAN control attributes. In future work, we will consider a principled method to learn the number and hierarchy of attributes in real-world data. In addition, improving the quality of samples with minority attributes is an important avenue for future research on unsupervised conditional GANs.

ACKNOWLEDGEMENT

This work was supported by the BK21 FOUR program of the Education and Research Program for Future ICT Pioneers, Seoul National University in 2021, Institute of Information & communications Technology Planning & Evaluation (IITP) grant funded by the Korea government(MSIT) [NO.2021-0-01343, Artificial Intelligence Graduate School Program (Seoul National University)], and AIR Lab (AI Research Lab) in Hyundai Motor Company through HMC-SNU AI Consortium Fund.

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

# A    ADDITIONAL RESULTS AND DISCUSSION

We also compared the proposed method with WGAN (Arjovsky et al., 2017) with the proposed method. We used the pre-activation of the penultimate layer of $D$ for the clustering metrics because WGAN has no encoder network. We could not measure ICFID of WGAN because it cannot perform unsupervised conditional generation. In addition to various datasets, we also used the 10x_73k dataset (Zheng et al., 2017), which consists of RNA transcript counts. From the results of the clustering performances on the 10x 73k dataset, we show that SLOGAN learns useful imbalanced attributes and can be helpful in the use of unlabeled biomedical data.

## A.1    PERFORMANCE COMPARISON ON BALANCED ATTRIBUTES

Table 6: Performance comparison on balanced attributes

| Dataset | Metric | WGAN | InfoGAN | DeLiGAN | DeLiGAN+ | ClusterGAN | SCGAN | CD-GAN | PGMGAN | **SLOGAN** |
|---|---|---|---|---|---|---|---|---|---|---|
| MNIST | NMI | 0.78±0.02 | 0.90±0.03 | 0.70±0.05 | 0.77±0.05 | 0.81±0.02 | 0.74±0.06 | 0.87±0.03 | 0.16±0.27 | **0.92±0.00** |
| | FID | 3.05±0.20 | 1.72±0.17 | 1.92±0.12 | 2.00±0.16 | 1.71±0.07 | 3.06±0.53 | 2.75±0.04 | 5.76±1.67 | **1.67±0.15** |
| | ICFID | N/A | 5.56±0.71 | 5.74±0.25 | 5.64±0.39 | 5.12±0.07 | 16.65±2.01 | 7.03±0.23 | 53.40±12.49 | **4.99±0.19** |
| FMNIST | NMI | 0.65±0.02 | 0.64±0.02 | 0.64±0.03 | 0.57±0.07 | 0.61±0.03 | 0.56±0.01 | 0.56±0.04 | 0.47±0.01 | **0.66±0.01** |
| | FID | 5.74±0.49 | 5.28±0.12 | 6.65±0.48 | 7.23±0.56 | 6.32±0.25 | **5.07±0.19** | 9.05±0.11 | 9.13±0.28 | 5.20±0.36 |
| | ICFID | N/A | 32.18±2.11 | 34.87±5.29 | 30.53±8.71 | 37.20±5.50 | 26.23±7.10 | 36.61±0.47 | 40.00±4.38 | **23.31±2.77** |
| CIFAR-2 | NMI | 0.14±0.02 | 0.05±0.03 | 0.15±0.13 | 0.12±0.12 | 0.34±0.02 | 0.00±0.00 | 0.38±0.01 | 0.67±0.00 | **0.78±0.03** |
| | FID | 29.54±0.59 | 58.84±13.11 | 338.97±70.85 | 116.95±19.42 | 36.28±1.12 | 39.44±1.72 | 34.45±0.74 | 29.49±0.51 | **28.99±0.36** |
| | ICFID | N/A | 91.97±14.21 | 361.66±71.28 | 153.19±17.71 | 47.02±1.85 | 71.54±5.41 | 43.98±1.47 | **35.67±0.61** | 35.68±0.51 |
| CIFAR-10 | NMI | 0.27±0.05 | 0.05±0.00 | 0.06±0.00 | 0.09±0.04 | 0.10±0.01 | 0.01±0.00 | 0.29±0.02 | **0.34±0.01** |
| | FID | 20.56±0.76 | 81.84±2.27 | 212.20±4.52 | 110.51±7.70 | 61.97±3.69 | 199.28±57.16 | 34.13±1.13 | 31.50±0.73 | 20.61±0.40 |
| | ICFID | N/A | 139.20±2.09 | 305.32±5.05 | 215.63±11.16 | 124.27±5.95 | 262.54±59.29 | 95.43±3.58 | 81.25±11.55 | **71.23±6.76** |

## A.2    PERFORMANCE COMPARISON ON IMALANCED ATTRIBUTES

Table 7: Performance comparison on imbalanced attributes

| Dataset | Metric | WGAN | InfoGAN | DeLiGAN | DeLiGAN+ | ClusterGAN | SCGAN | CD-GAN | PGMGAN | **SLOGAN** |
|---|---|---|---|---|---|---|---|---|---|---|
| MNIST-2 (7:3) | NMI | 0.90±0.03 | 0.28±0.19 | 0.90±0.04 | 0.48±0.09 | 0.27±0.19 | 0.67±0.11 | 0.41±0.03 | 0.79±0.21 | **0.92±0.05** |
| | FID | 4.27±0.19 | 4.92±0.85 | 4.21±0.84 | 4.63±2.02 | 4.25±1.06 | 4.34±0.73 | 4.67±1.92 | 8.90±14.82 | **4.02±0.86** |
| | ICFID | N/A | 36.35±10.65 | 25.34±1.72 | 26.61±1.49 | 25.41±1.02 | 16.47±1.51 | 26.71±2.47 | 14.82±9.16 | **5.91±1.06** |
| FMNIST-5 | NMI | 0.65±0.00 | 0.58±0.07 | **0.68±0.05** | 0.65±0.01 | 0.60±0.02 | 0.60±0.06 | 0.59±0.01 | 0.24±0.02 | 0.66±0.06 |
| | FID | 6.55±0.20 | 5.40±0.14 | 7.05±0.49 | 6.33±0.44 | 5.61±0.17 | **5.01±0.20** | 9.34±0.56 | 11.80±0.43 | 5.29±0.16 |
| | ICFID | N/A | 43.69±10.84 | 36.21±3.07 | 35.41±0.79 | 36.94±5.81 | 44.48±21.62 | 39.31±1.18 | 77.30±8.60 | **32.46±3.18** |
| 10x_73k | NMI | 0.22±0.04 | 0.42±0.06 | 0.61±0.01 | 0.60±0.01 | 0.66±0.02 | 0.47±0.02 | 0.68±0.03 | 0.33±0.07 | **0.76±0.02** |
| CIFAR-2 (7:3) | NMI | 0.09±0.07 | 0.05±0.01 | 0.00±0.00 | 0.03±0.03 | 0.22±0.02 | 0.00±0.00 | 0.22±0.03 | 0.42±0.03 | **0.69±0.02** |
| | FID | 29.16±0.90 | 51.30±2.53 | 131.73±50.98 | 115.19±17.95 | 36.62±2.16 | 45.28±1.81 | 36.40±1.01 | 29.76±1.65 | **29.09±0.73** |
| | ICFID | N/A | 88.49±6.85 | 186.31±28.31 | 173.81±18.29 | 75.52±4.82 | 88.58±4.57 | 76.91±1.07 | 57.06±3.31 | **45.83±3.03** |
| CIFAR-2 (9:1) | NMI | 0.04±0.04 | 0.00±0.00 | 0.02±0.02 | 0.09±0.11 | 0.02±0.01 | 0.00±0.00 | 0.05±0.03 | 0.16±0.03 | **0.38±0.01** |
| | FID | **29.37±0.53** | 60.76±8.97 | 129.50±25.33 | 139.75±47.13 | 41.69±0.83 | 50.45±1.56 | 38.15±2.70 | 30.23±1.31 | 29.47±1.53 |
| | ICFID | N/A | 138.24±10.23 | 205.26±10.93 | 196.00±17.86 | 133.31±2.03 | 123.35±6.56 | 128.46±3.03 | 101.68±3.87 | **86.75±1.87** |

## A.3    ABLATION STUDY

Table 8 shows the ablation study on SLOGAN trained with CIFAR-2 (7:3). $\pi_{y=0}$ and $\pi_{y=1}$ represent the mixing coefficients of the latent components that correspond to the frogs and planes, respectively, and the ground-truth of $\pi_{y=0}$ is 0.7.

**Factor analysis**    Rows 1-6 of Table 8 compare the performance depending on the factors affecting the performance of SLOGAN ($\boldsymbol{\mu}, \boldsymbol{\Sigma}, \boldsymbol{\rho}$ updates, and $\ell_{\text{U2C}}$). We confirmed that SLOGAN with all the factors demonstrated the highest performance. Among the parameters of the latent distribution, the $\boldsymbol{\mu}$ update leads to the highest performance improvement. The intra-cluster Fréchet inception distance (ICFID) of SLOGAN without $\boldsymbol{\rho}$ update (the 5th row of Table 8) indicates that SLOGAN outperformed existing unsupervised conditional GANs even when assuming a uniform distribution of the attributes.

**Loss for $\boldsymbol{\rho}$ update**    We do not use U2C loss $\ell_{\text{U2C}}$ to learn the mixing coefficient parameters $\boldsymbol{\rho}$. We construct U2C loss to approximate the negative mutual information $-I(C; \mathbf{x}_g)$ that can be

Table 8: Ablation study on CIFAR-2 (7:3)

| Ablation | $\pi_{y=0}$ (ground-truth: 0.7) | ICFID $\downarrow$ |
|---|---|---|
| **Factor analysis** | | |
| SLOGAN without $\boldsymbol{\mu}, \boldsymbol{\Sigma}, \boldsymbol{\rho}$ updates, $\ell_{\text{U2C}}$ | 0.50 | 84.44 |
| SLOGAN without $\boldsymbol{\mu}, \boldsymbol{\Sigma}, \boldsymbol{\rho}$ updates | 0.50 | 77.32 |
| SLOGAN without $\boldsymbol{\mu}$ update | 0.52 | 73.79 |
| SLOGAN without $\boldsymbol{\rho}$ update | 0.50 | 63.09 |
| SLOGAN without $\boldsymbol{\Sigma}$ update | 0.69 | 48.34 |
| SLOGAN without $\ell_{\text{U2C}}$ | 0.66 | 48.82 |
| **Implicit reparameterization** | | |
| DeLiGAN with $\ell_{\text{U2C}}$ | 0.50 | 60.51 |
| DeLiGAN with $\ell_{\text{U2C}}$ and implicit $\boldsymbol{\rho}$ update | 1.00 | 86.48 |
| **Loss for $\boldsymbol{\rho}$ update** | | |
| SLOGAN with $\ell_{\text{U2C}}$ for $\boldsymbol{\rho}$ update | 0.62 | 52.67 |
| **SimCLR analysis** | | |
| SLOGAN without SimCLR | 0.66 | 49.25 |
| SLOGAN without SimCLR on real data only | 0.67 | 48.41 |
| SLOGAN without SimCLR on both real and fake data | 0.69 | 47.93 |
| **Attribute manipulation** | | |
| SLOGAN with probe data | 0.71 | 44.97 |
| SLOGAN with probe data and mixup | 0.70 | 44.26 |
| SLOGAN | 0.69 | 45.83 |

Table 9: Effectiveness of U2C loss

| Dataset | Ablation | NMI $\uparrow$ | FID $\downarrow$ | ICFID $\downarrow$ |
|---|---|---|---|---|
| MNIST-2 | SLOGAN w/o $\ell_{\text{U2C}}$ | 0.25 | 4.62 | 9.43 |
| (7:3) | SLOGAN | **0.92** | **4.02** | **5.91** |
| FMNIST-5 | SLOGAN w/o $\ell_{\text{U2C}}$ | 0.14 | **5.27** | 43.15 |
| | SLOGAN | **0.66** | 5.29 | **32.46** |
| CIFAR-2 | SLOGAN w/o $\ell_{\text{U2C}}$ | 0.01 | 29.18 | 41.72 |
| | SLOGAN | **0.78** | **28.99** | **35.68** |
| CIFAR-2 | SLOGAN w/o $\ell_{\text{U2C}}$ | 0.08 | 30.34 | 48.82 |
| (7:3) | SLOGAN | **0.69** | **29.09** | **45.83** |
| CIFAR-10 | SLOGAN w/o $\ell_{\text{U2C}}$ | 0.08 | 20.91 | 78.26 |
| | SLOGAN | **0.34** | **20.61** | **71.23** |

decomposed into entropy and conditional entropy as follows:

$$\ell_{\text{U2C}}(\mathbf{z}) \approx -I(C; \mathbf{x}_g) = H(C|\mathbf{x}_g) - H(C) \tag{12}$$

The conditional entropy term reduces the uncertainty of the component from which the generated data are obtained. The entropy term promotes that component IDs are uniformly distributed. In terms of $\boldsymbol{\rho}$ update, the entropy term $H(C)$ drives $p(C)$ toward a discrete uniform distribution. Therefore, using $\ell_{\text{U2C}}$ for learning $\boldsymbol{\rho}$ pulls $\boldsymbol{\pi}$ to a discrete uniform distribution and can hinder the learned $\boldsymbol{\pi}$ from accurately estimating the imbalance ratio inherent in the data. In the 9th row of Table 8, we observed that the unsupervised conditional generation performance was undermined and the estimated imbalance ratio ($\pi_{y=0}$) was learned closer to a discrete uniform distribution (0.5) when $\ell_{\text{U2C}}$ was used for $\boldsymbol{\rho}$ update.

**SimCLR analysis**  For the colored image datasets, SLOGAN uses the SimCLR loss for the encoder with only fake (generated) data to further enhance the unsupervised conditional generation performance. The 10th to 12th rows of Table 8 show several ablation studies that analyzed the effect of SimCLR loss on SLOGAN. SLOGAN without SimCLR still showed at least approximately 35% performance improvement compared to the existing unsupervised conditional GANs (ICFID of

ClusterGAN: 75.52, CD-GAN: 76.91 in Table 2), even considering the fair computational cost and memory consumption. The SimCLR loss shows the highest performance improvement especially when applied only to fake data. SimCLR improved the performance by 7%.

**Attribute manipulation** As shown in the 13th and 14th rows of Table 8, probe data significantly improved the performance of SLOGAN on CIFAR-2 (7:3) with 10 probe data for each latent component. We also confirmed that the mixup applied to the probe data further enhanced the overall performance of our model. Figure 6 (a) shows the data generated from SLOGAN trained on CIFAR-2 (9:1) without the probe data. With extremely imbalanced attributes, SLOGAN mapped frog images with a white background onto the same component as airplanes in its latent space. When we use 10 probe data for each latent component, as shown in Figure 6 (b), frogs with a white background were generated from the same latent component as the other frog images.

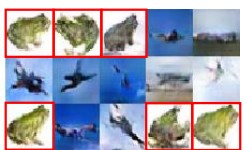 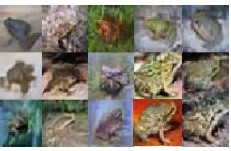 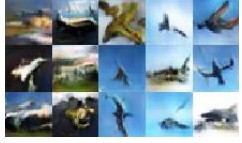 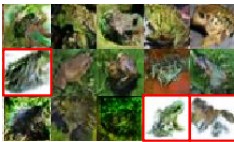

(a) SLOGAN (without probe data)          (b) SLOGAN with probe data & mixup

Figure 6: Effects of attribute manipulation on unsupervised conditional generation. Left and right images visualize generated images from different latent components. The red boxes indicate generated frog images with a white background.

**Feature scale** We introduced the feature scale $s$ described in Appendix D.1 to reinforce the discriminative power of U2C loss. For the MNIST-2 (7:3) dataset, $s$ was set to 4. Such a parameter configuration is justified by a greedy search in $[0.5, 1, 2, 4, 8]$. The performances of SLOGAN on MNIST-2 (7:3) with different feature scales are shown in Table 10.

Table 10: Ablation study on feature scale

| $s$ | 0.5 | 1 | 2 | 4 | 8 |
|---|---|---|---|---|---|
| ICFID $\downarrow$ | 14.18 | 17.03 | 6.65 | **5.91** | 33.98 |

Intuitively, increasing the feature scale $s$ makes the samples generated from the same component closer to each other in the embedding space. From these results, we observed that the optimal choice of the temperature factor enhances the discriminative power of U2C loss.

### A.4 STATISTICAL SIGNIFICANCE

We completed the statistical tests between SLOGAN and other methods in Tables 1 and 2. Since the results of the experiment could not satisfy normality and homogeneity of variance, we used the Wilcoxon rank sum test (Wilcoxon, 1992). When p-value $> 0.05$, we measured the effect size using Cohen's d (Cohen, 2013). We validated that all the experiments are statistically significant or showed large or medium effect sizes, with the exception of FID of InfoGAN vs. SLOGAN for FMNIST in Table 1, NMI of DeLiGAN+ vs. SLOGAN for FMNIST-5 in Table 2.

### A.5 INTERPOLATION IN LATENT SPACE

We also qualitatively show that the continuous nature of the prior distribution of SLOGAN makes superbly smooth interpolation possible in the latent space. In Figure 7, we visualize images generated from latent vectors obtained via interpolation among the mean vectors of the trained latent components. The generated images gradually changed to 3, 5, and 8 for the MNIST dataset, and t-shirt/top, pullover, and dress for the Fashion-MNIST dataset. In particular, we confirmed that the face images generated from the model trained with the CelebA data changed smoothly. The continuous attributes of real-world data are well mapped to the continuous latent space assumed by us, unlike most other methods using separated latent spaces induced via discrete latent codes.

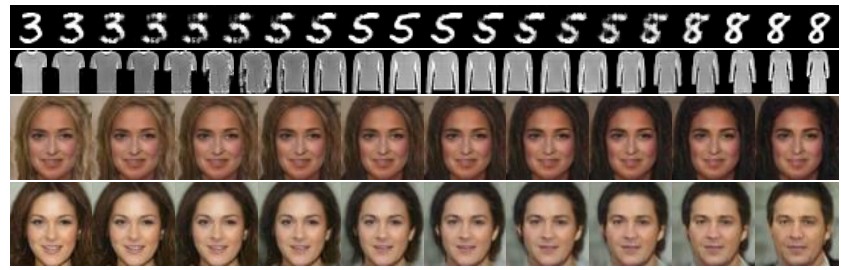

Figure 7: Interpolation in the latent space of the proposed method. For the MNIST and Fashion-MNIST datasets, we selected three mean vectors in the latent space and generated images from linearly interpolated latent vectors. For the CelebA dataset, we used 30 probe data and mixup for each latent component with attributes such as Black hair (3:1) and Male (1:1).

## A.6 BENEFITS OF ICFID

DeLiGAN and ClusterGAN trained on the MNIST-2 (7:3) exhibited comparable FIDs to SLOGAN (DeLiGAN: 4.21, ClusterGAN: 4.25, and SLOGAN: 4.02); however they showed ICFIDs approximately four times higher (DeLiGAN: 25.34, ClusterGAN: 25.61, and SLOGAN: 5.91). From the data generated from each latent component of DeLiGAN and ClusterGAN in Figure 8 (b) and (c), we confirm that the attributes were not learned well in the latent space of DeLiGAN. By contrast, from the data generated from each latent component of SLOGAN presented in Figure 8 (a), SLOGAN successfully learned the attributes in its latent space. This shows that ICFID is useful for evaluating the performance of unsupervised conditional generation. In addition, ICFID can evaluate the diversity of images generated from a discrete latent code or mode because ICFID is based on FID. As shown in Figure 9, when a mode collapse occurs, the diversity of samples decreases drastically, and DeLiGAN trained on the CIFAR-2 (7:3) shows approximately twice the ICFID than those of InfoGAN and ClusterGAN (DeLiGAN: 186.31, InfoGAN: 88.49, and ClusterGAN: 75:52).

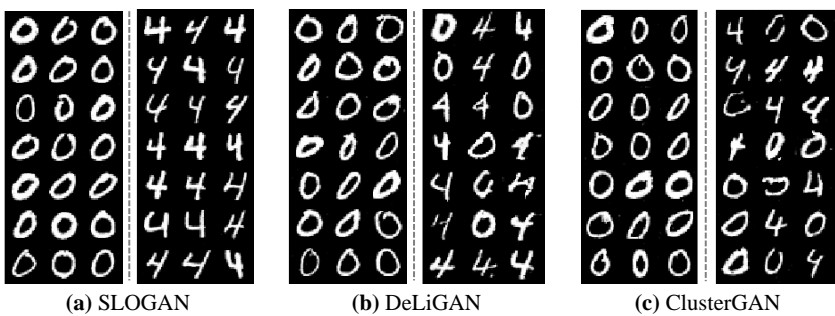

(a) SLOGAN      (b) DeLiGAN      (c) ClusterGAN

Figure 8: An example where ICFID is useful. The left and right images show generated images from each latent code of (a) SLOGAN, (b) DeLiGAN and (c) ClusterGAN trained on the MNIST-2 (7:3) dataset, respectively.

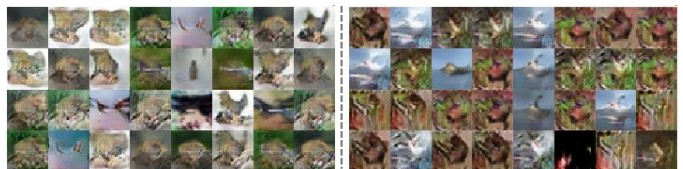

DeLiGAN

Figure 9: Another example where ICFID is useful. The left and right images show generated images from each latent component of DeLiGAN trained on the CIFAR-2 (7:3) dataset.

### A.7 QUALITATIVE RESULTS

**Synthetic data** Figures 10 and 11 shows the synthetic data and data generated by SLOGAN. Each color denotes generated data from each mixture component of the latent distribution. As the training progresses, each mixture component in the latent space is more strongly associated with each Gaussian distribution in the data space.

**Generated images and latent spaces** Figures 12, 13, 17, 18, 19, the left plot of Figure 15, and the upper plot of Figure 16 show the images generated from each latent component of SLOGAN trained on various datasets. Figure 14, the right plot of Figure 15, and the lower plot of Figure 16 visualize 1,000 latent vectors of SLOGAN trained on the MNIST, Fashion-MNIST, MNIST-2 (7:3), and FMNIST-5 datasets using 3D principal component analysis (PCA). Each color represents the component with the highest responsibility, and each image shows the generated image from the latent vector. As shown in Figure 14, similar attributes (e.g., 4, 7, and 9) are mapped to nearby components in the latent space.

**Comparisons with the most recent methods** We compare our method with the most recent methods such as CD-GAN (Pan et al., 2021) and PGMGAN (Armandpour et al., 2021) on the CIFAR-2 (7:3) dataset. From the results shown in Figure 20, we qualitatively confirm that SLOGAN learns imbalanced attributes of the dataset most robustly.

**Highly imbalanced multi-class data** We trained our method on highly imbalanced multi-class datasets by setting class 8 of the MNIST dataset to very low proportions of the other nine classes (e.g., 10:10:10:10:10:10:10:10:1:10 and 10:10:10:10:10:10:10:10:2:10). When class 8 is 0.1 fraction of the other nine classes, images of class 7 with a horizontal line outnumber images of class 8, and SLOGAN identifies 7 with a horizontal line as a more salient attribute than 8 as shown in the red boxes in Figure 21 (a). On the other hand, when class 8 is 0.2 fraction of the other nine classes, images of class 8 outnumber images of class 7 with a horizontal line. Therefore, SLOGAN successfully identifies 8 as a salient attribute as shown in the red box in Figure 21 (b).

**Qualitative analysis with various imbalance ratios** Figure 22 shows generated images from each latent component of SLOGAN trained on the AFHQ dataset. For various imbalance ratios of cats and dogs, we qualitatively analyze SLOGAN without using probe data. When the imbalance ratios are 1:1 and 1:2, SLOGAN identifies cat/dog as the most salient attribute and learned the attribute successfully as presented in Figure 22 (a) and (b). When the imbalance ratio is 1:5, SLOGAN discovers folded ears as the most salient attribute as shown in Figure 22 (c).

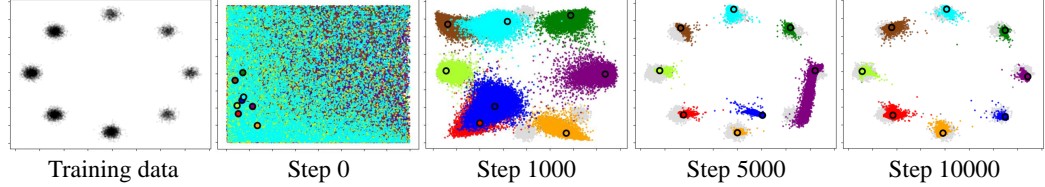

| Training data | Step 0 | Step 1000 | Step 5000 | Step 10000 |

Figure 10: Synthetic dataset and samples generated by SLOGAN at 0, 1000, 5000, and 10000 steps.

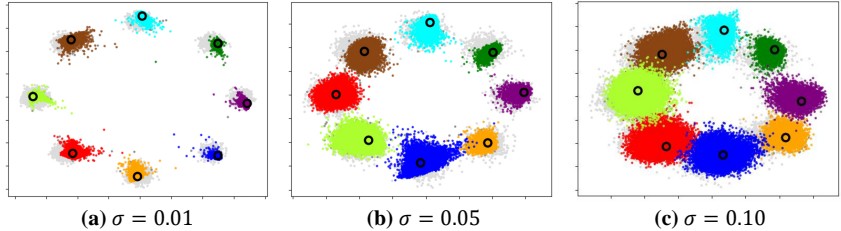

**(a)** $\sigma = 0.01$  **(b)** $\sigma = 0.05$  **(c)** $\sigma = 0.10$

Figure 11: Generated samples from each latent component of SLOGAN trained on the synthetic datasets with variances (a) $0.01I$, (b) $0.05I$, and (c) $0.10I$.

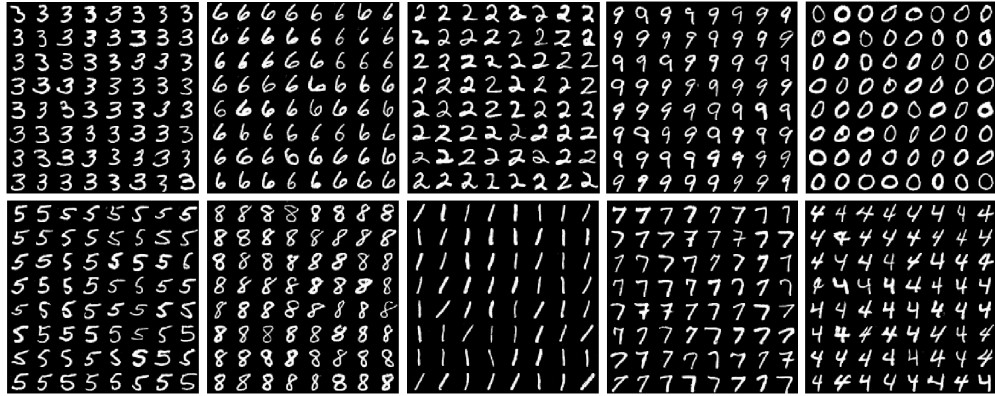

Figure 12: Generated images from each latent component of SLOGAN trained on the MNIST dataset.

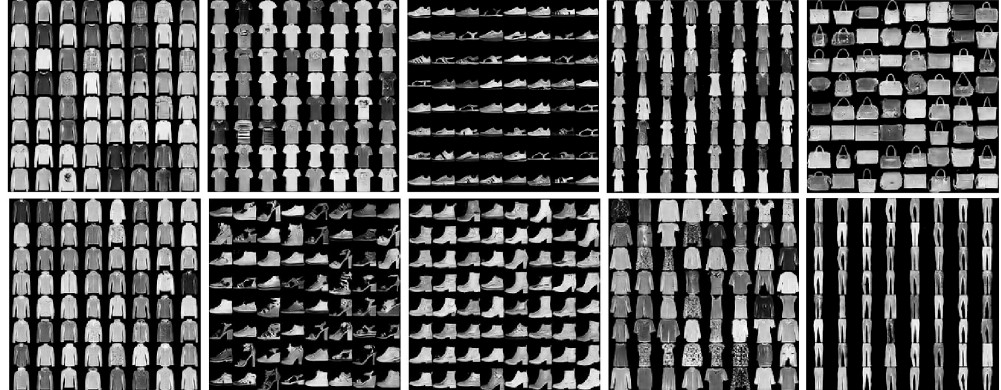

Figure 13: Generated images from each latent component of SLOGAN trained on the Fashion-MNIST dataset.

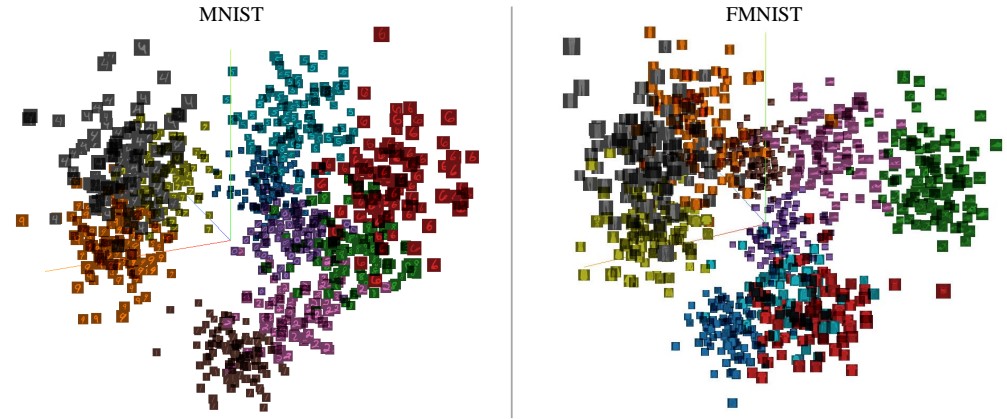

Figure 14: 3D PCA of the latent spaces of SLOGAN trained on the MNIST and Fashion-MNIST datasets.

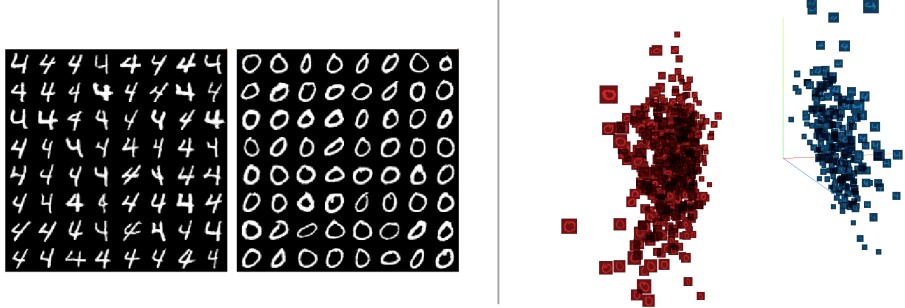

Figure 15: Generated images from each latent component and 3D PCA of the latent spaces of SLOGAN trained on the MNIST-2 (7:3) dataset.

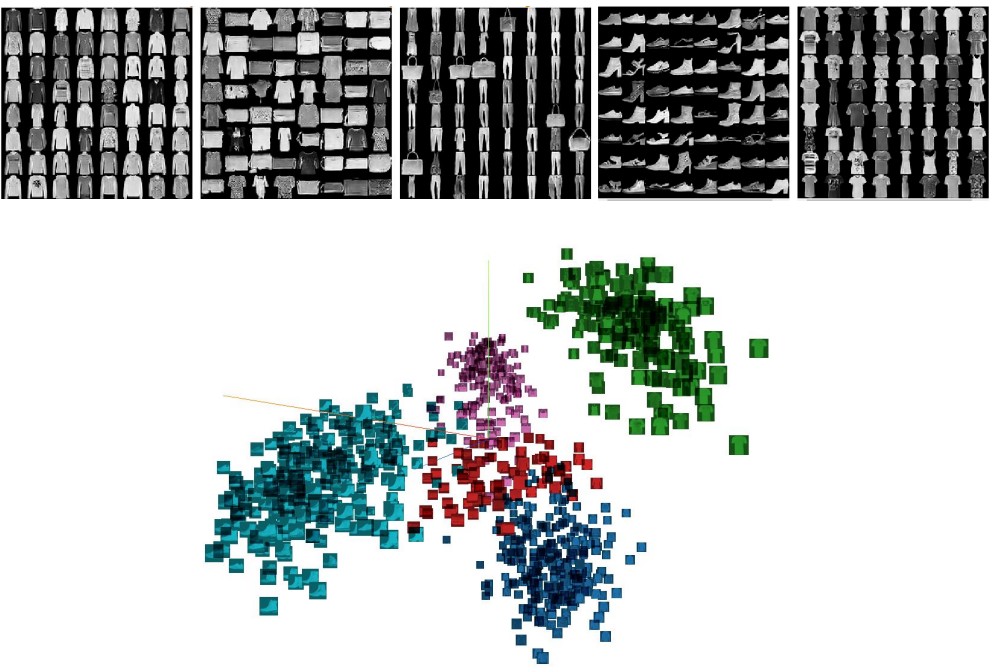

Figure 16: Generated images from each latent component and 3D PCA of the latent space of SLOGAN trained on the FMNIST-5 dataset.

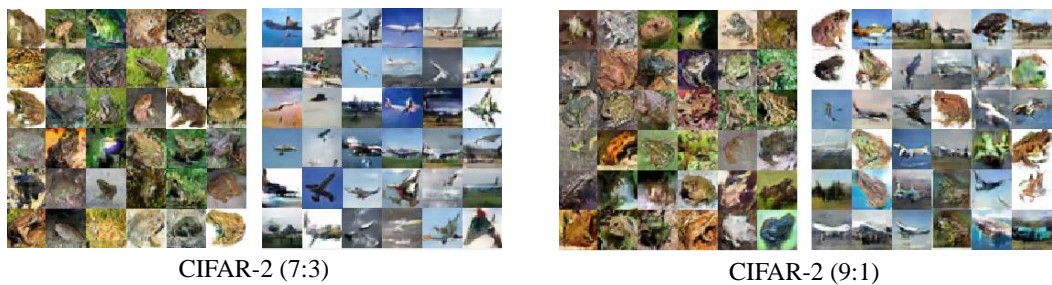

CIFAR-2 (7:3)                     CIFAR-2 (9:1)

Figure 17: Generated images from each latent component of SLOGAN trained on the CIFAR-2 (7:3) and CIFAR-2 (9:1) datasets.

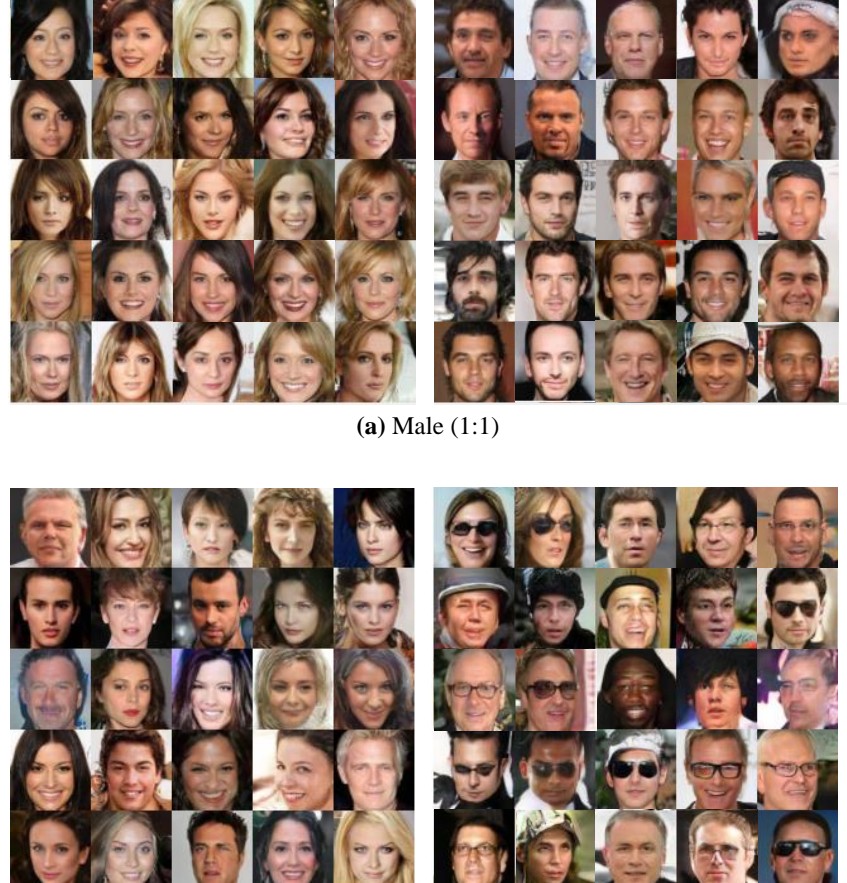

**(a)** Male (1:1)

**(b)** Eyeglasses (14:1)

Figure 18: Generated images from each latent component of SLOGAN trained on the CelebA dataset. We used 30 probe data ((a) Female vs. Male, or (b) Faces without eyeglasses vs. Faces with eyeglasses) and mixup for each component.

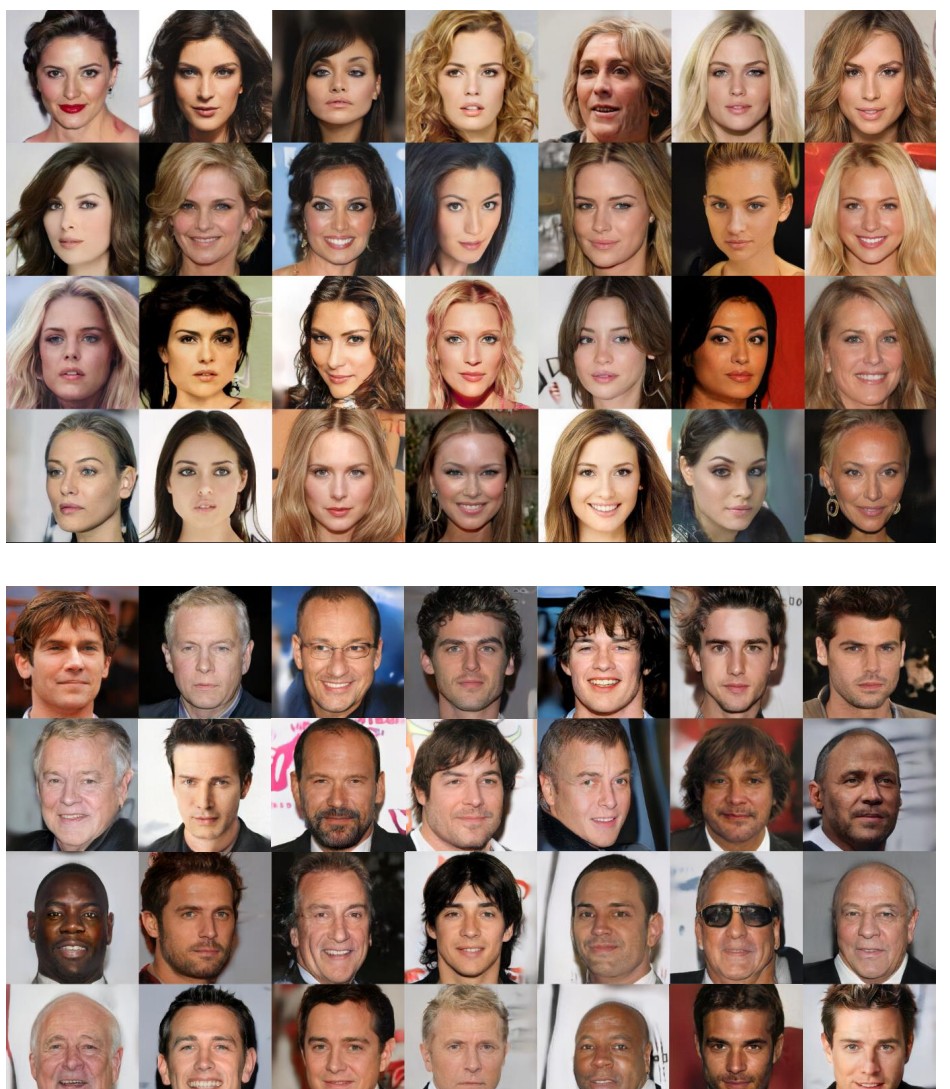

Figure 19: Generated images from each latent component of SLOGAN trained on the CelebA-HQ (256×256) dataset. We used 30 probe data (Female vs. Male) and mixup for each component.

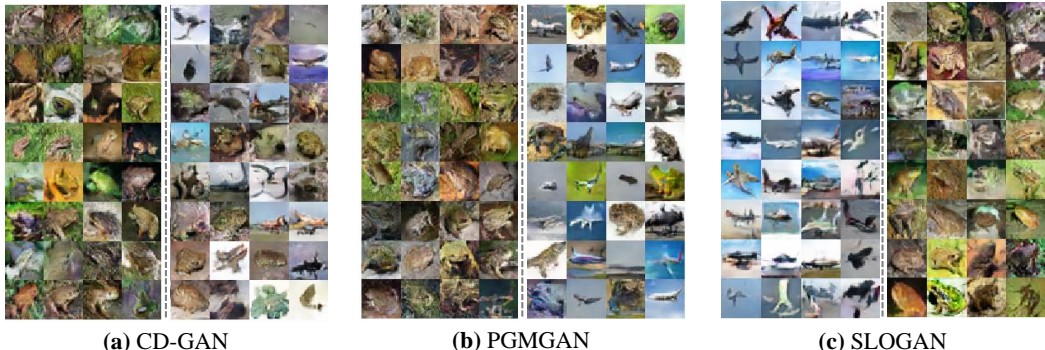

**(a)** CD-GAN      **(b)** PGMGAN      **(c)** SLOGAN

Figure 20: Generated images from the most recent methods including (a) CD-GAN, (b) PGMGAN, and (c) SLOGAN trained on the CIFAR-2 (7:3) dataset.

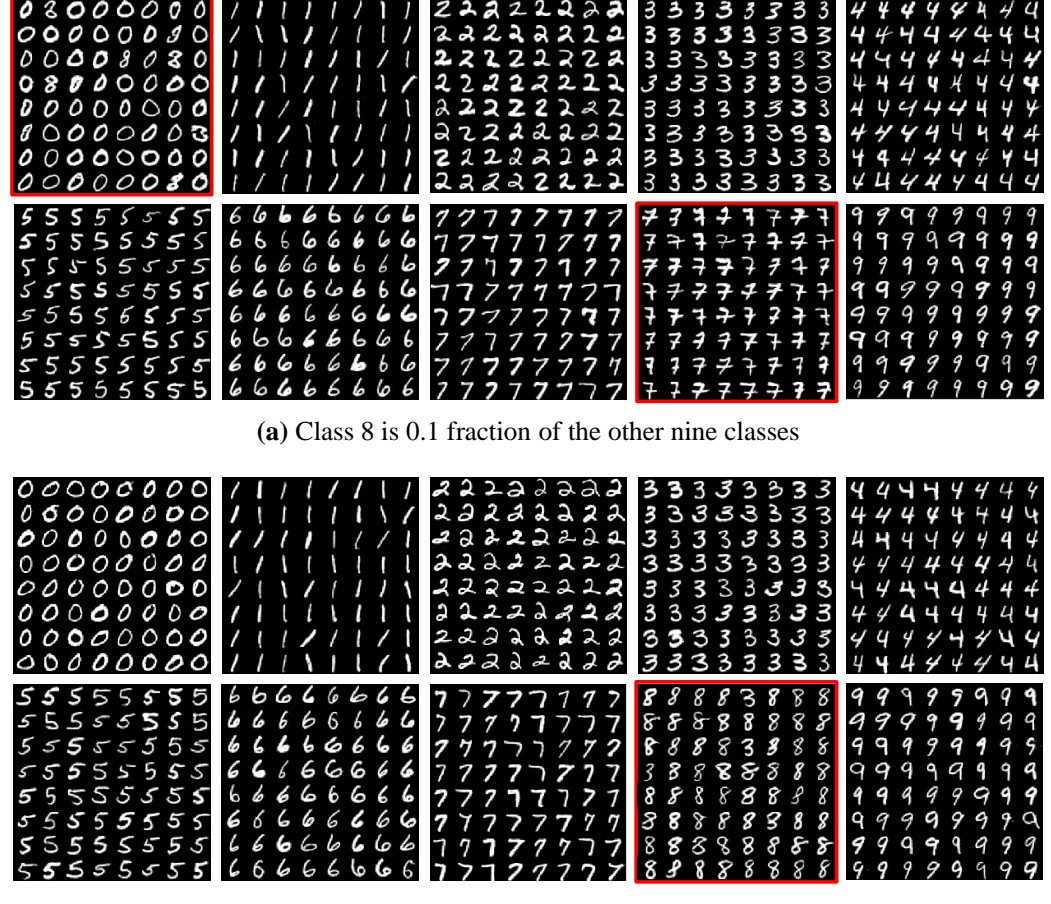

**(a)** Class 8 is 0.1 fraction of the other nine classes

**(b)** Class 8 is 0.2 fraction of the other nine classes

Figure 21: Generated images from each latent component of SLOGAN trained on the MNIST dataset where class 8 is very low fraction of the other nine classes.

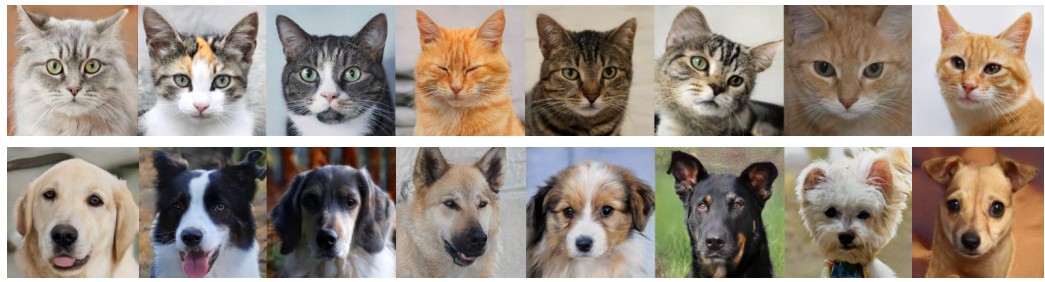

**(a)** Cat:Dog = 1:1

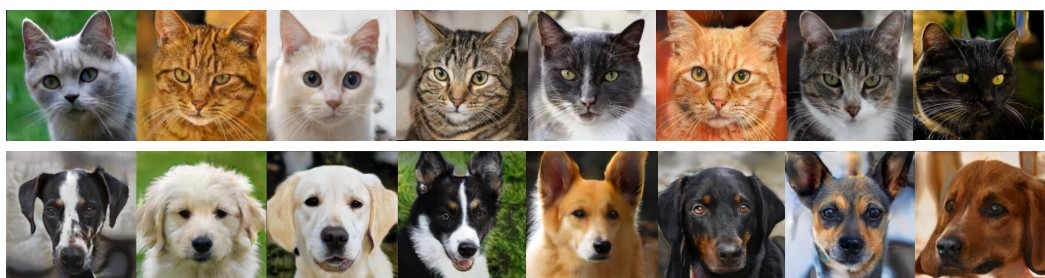

**(b)** Cat:Dog = 1:2

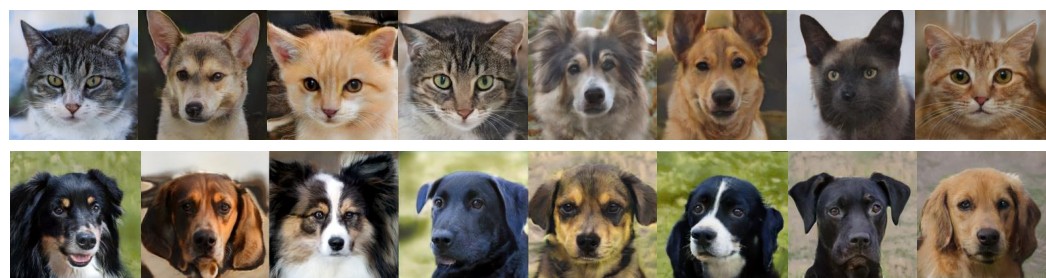

**(c)** Cat:Dog = 1:5

Figure 22: Generated images from each latent component of SLOGAN trained on Cats and Dogs of the AFHQ (256×256) dataset with various imbalance ratios.

# B  ADDITIONAL BACKGROUND

Our work is closely related to Stein's lemma and the reparameterization trick for Gaussian mixture and also related to several topics of GAN studies such as representation learning, supervised/unsupervised conditional generation, and Gaussian mixture prior.

## B.1  SUPERVISED CONDITIONAL GENERATION

Conditional GANs including ACGAN (Odena et al., 2017), projection discriminator (Miyato & Koyama, 2018), and ContraGAN (Kang & Park, 2020) have led to state-of-the-art performances in conditional image generation. However, these conditional GANs are trained with supervision and require a large amount of labeled data.

## B.2  UNSUPERVISED CONDITIONAL GENERATION

InfoGAN (Chen et al., 2016) introduces latent codes composed of categorical and continuous variables and derives the lower bound of mutual information between the latent codes and representations. By maximizing the lower bound, InfoGAN learns disentangled latent variables. ClusterGAN (Mukherjee et al., 2019) assumes a discrete-continuous Gaussian prior wherein discrete variables are defined as a one-hot vector and continuous variables are sampled from a Gaussian distribution. The latent distribution is clustered in an unsupervised manner through reconstruction loses for discrete and continuous variables. CD-GAN (Pan et al., 2021) is similar to ClusterGAN, but it uses contrastive loss to disentangle attributes. These methods assume uniform distribution of the attributes, and if the imbalance ratio is unknown, these models cannot thoroughly learn imbalanced attributes.

Recently, unsupervised conditional GANs which do not assume uniform distribution of the attributes have been proposed. Self-conditioned GAN (Liu et al., 2020) performs unsupervised conditional generation by using clustering of discriminator features as labels. However, it has no loss to facilitate clustering of discriminator features, and the performance seems to be largely influenced by the architecture of the discriminator. PGMGAN (Armandpour et al., 2021) applies a contrastive clustering method named SCAN (Van Gansbeke et al., 2020) to perform unsupervised conditional generation. The pretrained space partitioner and a penalization loss function are used to encourage the generator to generate images with salient attributes. To avoid degenerated clusters where one partition contains most of the data, PGMGAN uses an entropy term that regularizes the average cluster probability vector to a uniform distribution. However, when learning PGMGAN on datasets with imbalanced attributes, it is difficult to adjust the coefficient of the regularizer because the performance seems to be sensitive to the strength of the regularizer in practice (e.g., the results on CIFAR-2 (7:3) and (9:1) in Table 7). In addition, clustering results does not seem to be reliable when the space partitioner is trained on datasets where transformations are not available (e.g., the result on 10x_73k in Table 7).

NEMGAN (Mishra et al., 2020) argues that it considers the imbalance problem of attributes. However, it assumes that some labeled samples are provided, which is unrealistic in real-world scenarios. In addition, labeled samples should have the same imbalance ratio as the training data to estimate the imbalance ratio. If these samples are not given, NEMGAN is the same method as ClusterGAN. On the other hand, SLOGAN does not require labeled samples to learn imbalanced attributes. Even when a small amount of probe data is used to learn specific attributes, imbalanced attributes can be learned with balanced probe data (e.g., 30 male and 30 female faces on the CelebA-HQ dataset).

## B.3  REPRESENTATION LEARNING

Representation learning refers to the discovery of meaningful semantics in datasets (Bengio et al., 2013). Perceptually salient and semantically meaningful representations can induce better performance in downstream tasks. In particular, Representation learning in generative models enables the learned latent representations to be semantically meaningful without labels in the training data. BiGAN (Donahue et al., 2017), ALI (Dumoulin et al., 2017), and their variants (Jaiswal et al., 2018; Belghazi et al., 2018) are similar to our study in that they add an encoder network to the original GAN framework. The additional encoder may serve as a inversion mapping of the generator, and learned feature representation is useful for supervised tasks. However, the learned generator cannot perform conditional generation without supervision.

## C  PROOFS

### C.1  GRADIENT IDENTITY FOR $\boldsymbol{\mu}_c$

**Theorem 1.** *Given an expected loss of the generator $\mathcal{L}$ and a loss function for a sample $\ell(\cdot) : \mathbb{R}^{d_z} \mapsto \mathbb{R}$, we assume $\ell$ is continuously differentiable. Then, the following identity holds:*

$$\nabla_{\boldsymbol{\mu}_c}\mathcal{L} = \mathbb{E}_q\left[\delta(\mathbf{z})_c\pi_c\nabla_{\mathbf{z}}\ell(\mathbf{z})\right] \tag{13}$$

*Proof.* To prove Theorem 1, the following lemma (Bonnet's theorem) is introduced.

**Lemma 2.** *Let $h(\mathbf{z}) : \mathbb{R}^d \mapsto \mathbb{R}$ be continuously differentiable. $q(\mathbf{z})$ is a multivariate Gaussian distribution $\mathcal{N}(\mathbf{z}; \boldsymbol{\mu}, \boldsymbol{\Sigma})$. Then, the following identity holds:*

$$\nabla_{\boldsymbol{\mu}}\mathbb{E}_{q(\mathbf{z})}\left[h(\mathbf{z})\right] = \mathbb{E}_{q(\mathbf{z})}\left[\nabla_{\mathbf{z}}h(\mathbf{z})\right] \tag{14}$$

The proof of Lemma 2 is described by Theorem 3 of Lin et al. (2019). Using Lemma 2, we show that

$$\nabla_{\boldsymbol{\mu}_c}\mathcal{L} = \nabla_{\boldsymbol{\mu}_c}\mathbb{E}_{q(\mathbf{z})}\left[\ell(\mathbf{z})\right] = \nabla_{\boldsymbol{\mu}_c}\sum_{i=1}^{K}p(i)\mathbb{E}_{q(\mathbf{z}|i)}\left[\ell(\mathbf{z})\right] \tag{15}$$

$$= p(c)\nabla_{\boldsymbol{\mu}_c}\mathbb{E}_{q(\mathbf{z}|c)}\left[\ell(\mathbf{z})\right] = p(c)\mathbb{E}_{q(\mathbf{z}|c)}\left[\nabla_{\mathbf{z}}\ell(\mathbf{z})\right] \tag{16}$$

$$= \int q(\mathbf{z}|c)p(c)\nabla_{\mathbf{z}}\ell(\mathbf{z})d\mathbf{z} \tag{17}$$

$$= \int q(\mathbf{z})\frac{q(\mathbf{z}|c)p(c)}{q(\mathbf{z})}\nabla_{\mathbf{z}}\ell(\mathbf{z})d\mathbf{z} \tag{18}$$

$$= \int q(\mathbf{z})\delta(\mathbf{z})_c\pi_c\nabla_{\mathbf{z}}\ell(\mathbf{z})d\mathbf{z} \tag{19}$$

$$= \mathbb{E}_{q(\mathbf{z})}\left[\delta(\mathbf{z})_c\pi_c\nabla_{\mathbf{z}}\ell(\mathbf{z})\right] \tag{20}$$

$\square$

### C.2  FIRST-ORDER GRADIENT IDENTITY FOR $\boldsymbol{\Sigma}_c$

**Theorem 2.** *With the same assumptions from Theorem 1, we have the following gradient identity:*

$$\nabla_{\Sigma_c}\mathcal{L} = \frac{1}{2}\mathbb{E}_q\left[\delta(\mathbf{z})_c\pi_c\Sigma_c^{-1}\left(\mathbf{z} - \boldsymbol{\mu}_c\right)\nabla_{\mathbf{z}}^T\ell(\mathbf{z})\right] \tag{21}$$

*Proof.* In order to prove Theorem 2, we introduce the following lemma (Price's theorem).

**Lemma 3.** *Let $h(\mathbf{z}) : \mathbb{R}^d \mapsto \mathbb{R}$ and its derivative $\nabla h(\mathbf{z})$ be continuously differentiable. We further assume that $\mathbb{E}\left[h(\mathbf{z})\right]$ is well-defined. Then, the following identity holds:*

$$\nabla_{\boldsymbol{\Sigma}}\mathbb{E}_{q(\mathbf{z})}\left[h(\mathbf{z})\right] = \frac{1}{2}\mathbb{E}_{q(\mathbf{z})}\left[\boldsymbol{\Sigma}^{-1}(\mathbf{z} - \boldsymbol{\mu})\nabla_{\mathbf{z}}^T h(\mathbf{z})\right] = \frac{1}{2}\mathbb{E}_{q(\mathbf{z})}\left[\nabla_{\mathbf{z}}^2 h(\mathbf{z})\right] \tag{22}$$

The proof of Lemma 3 is presented in Theorem 4 of Lin et al. (2019). The rest of the proof is similar with the proof of Theorem 1. Using the first-order gradient identity of Lemma 3, we get

$$\nabla_{\Sigma_c}\mathcal{L} = \nabla_{\Sigma_c}\mathbb{E}_{q(\mathbf{z})}\left[\ell(\mathbf{z})\right] = \nabla_{\Sigma_c}\sum_{i=1}^{K}p(i)\mathbb{E}_{q(\mathbf{z}|i)}\left[\ell(\mathbf{z})\right] \tag{23}$$

$$= p(c)\nabla_{\Sigma_c}\mathbb{E}_{q(\mathbf{z}|c)}\left[\ell(\mathbf{z})\right] = \frac{1}{2}p(c)\mathbb{E}_{q(\mathbf{z}|c)}\left[\Sigma_c^{-1}(\mathbf{z} - \boldsymbol{\mu}_c)\nabla_{\mathbf{z}}^T\ell(\mathbf{z})\right] \tag{24}$$

$$= \frac{1}{2}\int q(\mathbf{z}|c)p(c)\Sigma_c^{-1}(\mathbf{z} - \boldsymbol{\mu}_c)\nabla_{\mathbf{z}}^T\ell(\mathbf{z})d\mathbf{z} \tag{25}$$

$$= \frac{1}{2}\int q(\mathbf{z})\frac{q(\mathbf{z}|c)p(c)}{q(\mathbf{z})}\Sigma_c^{-1}(\mathbf{z} - \boldsymbol{\mu}_c)\nabla_{\mathbf{z}}^T\ell(\mathbf{z})d\mathbf{z} \tag{26}$$

$$= \frac{1}{2} \int q(\mathbf{z}) \delta(\mathbf{z})_c \pi_c \Sigma_c^{-1} (\mathbf{z} - \boldsymbol{\mu}_c) \nabla_{\mathbf{z}}^T \ell(\mathbf{z}) d\mathbf{z} \tag{27}$$

$$= \frac{1}{2} \mathbb{E}_{q(\mathbf{z})} \left[ \delta(\mathbf{z})_c \pi_c \Sigma_c^{-1} (\mathbf{z} - \boldsymbol{\mu}_c) \nabla_{\mathbf{z}}^T \ell(\mathbf{z}) \right] \tag{28}$$

$\square$

### C.3 Ensuring Positive-definiteness of $\Sigma_c$

**Theorem 3.** *The updated covariance matrix $\Sigma_c' = \Sigma_c + \gamma \Delta \Sigma_c'$ with the modified update rule specified in Equation 6 is positive-definite if $\Sigma_c$ is positive-definite.*

*Proof.* Because $\Sigma_c$ is symmetric and positive-definite, we can decompose $\Sigma_c = LL^T$ using Cholesky decomposition, where L is the lower triangular matrix. Then, we can prove the positive-definiteness of the updated covariance matrix as follows:

$$\Sigma_c' = \Sigma_c + \gamma \Delta \Sigma_c' \tag{29}$$

$$= \Sigma_c + \gamma (\Delta \Sigma_c + \frac{\gamma}{2} \Delta \Sigma_c \Sigma_c^{-1} \Delta \Sigma_c) \tag{30}$$

$$= \Sigma_c + \gamma \Delta \Sigma_c + \frac{\gamma^2}{2} \Delta \Sigma_c \Sigma_c^{-1} \Delta \Sigma_c \tag{31}$$

$$= \frac{1}{2} \left( \Sigma_c + (L + \gamma \Delta \Sigma_c L^{-T})(L^T + \gamma L^{-1} \Delta \Sigma_c) \right) \tag{32}$$

$$\tag{33}$$

Let us define $U := L^T + \gamma L^{-1} \Delta \Sigma_c$. Then, we have the following:

$$\Sigma_c' = \frac{1}{2} \left( \Sigma_c + U^T U \right) \succ 0 \tag{34}$$

where both $\Sigma_c$ and $U^T U$ are positive-definite, concluding the proof. $\square$

### C.4 Gradient Identity for $\rho_c$

**Theorem 4.** *Let $\rho_c$ be a mixing coefficient parameter, and the following gradient identity holds:*

$$\nabla_{\rho_c} \mathcal{L} = \mathbb{E}_q \left[ \pi_c \left( \delta(\mathbf{z})_c - 1 \right) \ell(\mathbf{z}) \right] \tag{35}$$

*Proof.* The gradient of the latent distribution with respect to the mixing coefficient parameter is derived as follows:

$$\nabla_{\rho_c} q(\mathbf{z}) = \nabla_{\rho_c} \sum_{i=1}^{K} \mathrm{softmax}(\rho_i) q(\mathbf{z}|i) = \pi_c \left( q(\mathbf{z}|c) - \sum_{i=1}^{K} \pi_i q(\mathbf{z}|i) \right) \tag{36}$$

where $\mathrm{softmax}(\cdot)$ denotes the softmax function (e.g., $p(i) = \pi_i = \mathrm{softmax}(\rho_i)$). Using the above equation, we have

$$\nabla_{\rho_c} \mathcal{L} = \nabla_{\rho_c} \mathbb{E}_{q(\mathbf{z})} \left[ \ell(\mathbf{z}) \right] = \int \nabla_{\rho_c} q(\mathbf{z}) \ell(\mathbf{z}) d\mathbf{z} \tag{37}$$

$$= \int \pi_c \left( q(\mathbf{z}|c) - \sum_{i=1}^{K} \pi_i q(\mathbf{z}|i) \right) \ell(\mathbf{z}) d\mathbf{z} \tag{38}$$

$$= \int \pi_c q(\mathbf{z}) \left( \frac{q(\mathbf{z}|c)}{q(\mathbf{z})} - \sum_{i=1}^{K} \pi_i \frac{q(\mathbf{z}|i)}{q(\mathbf{z})} \right) \ell(\mathbf{z}) d\mathbf{z} \tag{39}$$

$$= \int q(\mathbf{z}) \pi_c \left( \delta(\mathbf{z})_c - \sum_{i=1}^{K} \pi_i \delta(\mathbf{z})_i \right) \ell(\mathbf{z}) d\mathbf{z} \tag{40}$$

Here, $\sum_{i=1}^{K} \pi_i \delta(\mathbf{z})_i = \sum_{i=1}^{K} \frac{p(i)q(\mathbf{z}|i)}{q(\mathbf{z})} = 1$. Plugging this back into Equation 40, we obtain

$$\nabla_{\rho_c}\mathcal{L} = \int q(\mathbf{z})\pi_c\left(\delta(\mathbf{z})_c - 1\right)\ell(\mathbf{z})d\mathbf{z} \tag{41}$$

$$= \mathbb{E}_{q(\mathbf{z})}\left[\pi_c\left(\delta(\mathbf{z})_c - 1\right)\ell(\mathbf{z})\right] \tag{42}$$

$\square$

## C.5 Second-order Gradient Identity for $\boldsymbol{\Sigma}_c$

We consider a generator $\sigma(G(\mathbf{z})) = \sigma(G) = \boldsymbol{\sigma}_G$, a discriminator $D(\boldsymbol{\sigma}_G) = D$ with the piecewise-linear activation functions (e.g. ReLU or LeakyReLU) except the activation function of the output layer of the generator which is the hyperbolic tangent function $\sigma(x) = (e^x - e^{-x})/(e^x + e^{-x})$, and a Wasserstein adversarial loss function for each latent vector $\ell(\mathbf{z}) = -D(\sigma(G(\mathbf{z}))) = \ell$. We note that the following holds except on a set of zero Lebesgue measure because the piecewise-linear activation functions are linear except where they switch:

$$\frac{\partial^2 D}{\partial \boldsymbol{\sigma}_G{}^2} = 0, \quad \frac{\partial^2 G}{\partial \mathbf{z}^2} = 0 \tag{43}$$

We provide the second-order gradient identity of the Wasserstein GAN loss for the covariance matrix $\boldsymbol{\Sigma}_c$ that does not compute second-order derivatives. However, it is impractical and not included in our method because of the excessive computational cost of Jacobian matrices.

**Theorem 5.** *Given the Wasserstein GAN loss for the generator $\mathcal{L}$ and a loss function for a sample $\ell(\mathbf{z}) : \mathbb{R}^d \mapsto \mathbb{R}$, we assume $\ell$ and its derivative $\nabla\ell(\mathbf{z})$ are continuously differentiable. We further assume that $\mathbb{E}\left[\ell(\mathbf{z})\right]$ is well-defined. Then, the following identity holds:*

$$\nabla_{\Sigma_c}\mathcal{L} = \nabla_{\Sigma_c}\mathbb{E}_{q(\mathbf{z})}\left[\ell(\mathbf{z})\right] = -\mathbb{E}_{q(\mathbf{z})}\left[\pi_c\delta(\mathbf{z})_c \nabla_{\mathbf{z}}^T G(\mathbf{z})\operatorname{diag}\left(\nabla_{\boldsymbol{\sigma}_G}D \odot (\boldsymbol{\sigma}_G^{\circ 3} - \boldsymbol{\sigma}_G)\right)\nabla_{\mathbf{z}}G(\mathbf{z})\right]$$

*where $\odot$ denotes element-wise multiplication of vectors and $\mathbf{x}^{\circ i}$ denotes the i-th Hadamard (element-wise) power of a vector $\mathbf{x}$.*

*Proof.* From the second-order identity of Lemma 3, we have the following:

$$\nabla_{\Sigma_c}\mathcal{L} = \nabla_{\Sigma_c}\mathbb{E}_{q(\mathbf{z})}\left[\ell(\mathbf{z})\right] = \nabla_{\Sigma_c}\sum_{i=1}^{K} p(i)\mathbb{E}_{q(\mathbf{z}|i)}\left[\ell(\mathbf{z})\right] \tag{44}$$

$$= p(c)\nabla_{\Sigma_c}\mathbb{E}_{q(\mathbf{z}|c)}\left[\ell(\mathbf{z})\right] = \frac{1}{2}p(c)\mathbb{E}_{q(\mathbf{z}|c)}\left[\nabla_{\mathbf{z}}^2\ell(\mathbf{z})\right] \tag{45}$$

$$= \frac{1}{2}\mathbb{E}_{q(\mathbf{z})}\left[\frac{q(\mathbf{z}|c)p(c)}{q(\mathbf{z})}\nabla_{\mathbf{z}}^2\ell(\mathbf{z})\right] \tag{46}$$

$$= \frac{1}{2}\mathbb{E}_{q(\mathbf{z})}\left[\pi_c\delta(\mathbf{z})_c \nabla_{\mathbf{z}}^2\ell(\mathbf{z})\right] \tag{47}$$

The Hessian of the sample loss with respect to $\mathbf{z}$ is given as follows:

$$\nabla_{\mathbf{z}}^2\ell(\mathbf{z}) = \nabla_{\mathbf{z}}\left(-\nabla_{\mathbf{z}}^T D\right) = \nabla_{\mathbf{z}}\left(-(\nabla_G\boldsymbol{\sigma}_G\nabla_{\mathbf{z}}G)^T\nabla_{\boldsymbol{\sigma}_G}D\right)^T \tag{48}$$

$$= \nabla_{\mathbf{z}}\left(-(\nabla_{\boldsymbol{\sigma}_G}D)^T\nabla_G\boldsymbol{\sigma}_G\nabla_{\mathbf{z}}G\right) \tag{49}$$

$$= -\frac{\partial}{\partial\mathbf{z}}\left[\frac{\partial D}{\partial(\sigma_G)_1}(\sigma'_G)_1 \quad \cdots \quad \frac{\partial D}{\partial(\sigma_G)_{d_x}}(\sigma'_G)_{d_x}\right]\begin{bmatrix}\frac{\partial G_1}{\partial z_1} & \cdots & \frac{\partial G_1}{\partial z_{d_z}} \\ \vdots & \ddots & \vdots \\ \frac{\partial G_{d_x}}{\partial z_1} & \cdots & \frac{\partial G_{d_x}}{\partial z_{d_z}}\end{bmatrix} \tag{50}$$

$$= -\frac{\partial}{\partial\mathbf{z}}\left[\sum_{i=1}^{d_x}\frac{\partial D}{\partial(\sigma_G)_i}(\sigma'_G)_i\frac{\partial G_i}{\partial z_1} \quad \cdots \quad \sum_{i=1}^{d_x}\frac{\partial D}{\partial(\sigma_G)_i}(\sigma'_G)_i\frac{\partial G_i}{\partial z_{d_z}}\right] \tag{51}$$

$$= -\begin{bmatrix}\frac{\partial}{\partial z_1}\left(\sum_{i=1}^{d_x}\frac{\partial D}{\partial(\sigma_G)_i}(\sigma'_G)_i\frac{\partial G_i}{\partial z_1}\right) & \cdots & \frac{\partial}{\partial z_1}\left(\sum_{i=1}^{d_x}\frac{\partial D}{\partial(\sigma_G)_i}(\sigma'_G)_i\frac{\partial G_i}{\partial z_{d_z}}\right) \\ \vdots & \ddots & \vdots \\ \frac{\partial}{\partial z_{d_z}}\left(\sum_{i=1}^{d_x}\frac{\partial D}{\partial(\sigma_G)_i}(\sigma'_G)_i\frac{\partial G_i}{\partial z_1}\right) & \cdots & \frac{\partial}{\partial z_{d_z}}\left(\sum_{i=1}^{d_x}\frac{\partial D}{\partial(\sigma_G)_i}(\sigma'_G)_i\frac{\partial G_i}{\partial z_{d_z}}\right)\end{bmatrix} \tag{52}$$

where $\boldsymbol{\sigma}'_G = \nabla_G \boldsymbol{\sigma}_G$. To simplify an element $(\nabla^2_{\mathbf{z}} \ell(\mathbf{z}))_{jk} = -\frac{\partial}{\partial z_j} \left( \sum_{i=1}^{d_x} \frac{\partial D}{\partial (\sigma_G)_i} (\sigma'_G)_i \frac{\partial G_i}{\partial z_k} \right)$ for arbitrary $j, k \in \{1, ..., d_z\}$, we have

$$\frac{\partial^2 D}{\partial z_j \partial (\sigma_G)_i} = \frac{\partial}{\partial (\sigma_G)_i} \frac{\partial D}{\partial z_j} = \frac{\partial}{\partial (\sigma_G)_i} \left( \sum_{l=1}^{d_x} \frac{\partial D}{\partial (\sigma_G)_l} \frac{\partial (\sigma_G)_l}{\partial z_j} \right) \tag{53}$$

$$= \frac{\partial}{\partial (\sigma_G)_i} \left( \sum_{l=1}^{d_x} \frac{\partial D}{\partial (\sigma_G)_l} (\sigma'_G)_l \frac{\partial G_l}{\partial z_j} \right) \tag{54}$$

$$= \sum_{l=1}^{d_x} \frac{\cancel{\partial^2 D}}{\partial (\sigma_G)_i \partial (\sigma_G)_l} (\sigma'_G)_l \frac{\partial G_l}{\partial z_j} + \sum_{l=1}^{d_x} \frac{\partial D}{\partial (\sigma_G)_l} \frac{\partial (\sigma'_G)_l}{\partial (\sigma_G)_i} \frac{\partial G_l}{\partial z_j} \tag{55}$$

$$+ \sum_{l=1}^{d_x} \frac{\partial D}{\partial (\sigma_G)_l} (\sigma'_G)_l \frac{\partial^2 G_l}{\partial (\sigma_G)_i \partial z_j} \tag{56}$$

$$= \frac{\partial D}{\partial (\sigma_G)_i} (-2(\sigma_G)_i) \frac{\partial G_i}{\partial z_j} + \frac{\partial D}{\partial (\sigma_G)_i} (\sigma'_G)_i \frac{\partial}{\partial z_j} \left( \frac{1}{(\sigma'_G)_i} \right) \tag{57}$$

$$= \frac{\partial D}{\partial (\sigma_G)_i} (-2(\sigma_G)_i) \frac{\partial G_i}{\partial z_j} + \frac{\partial D}{\partial (\sigma_G)_i} \cancel{(\sigma'_G)_i} \frac{2(\sigma_G)_i}{\cancel{(\sigma'_G)_i}} \frac{\partial G_i}{\partial z_j} \tag{58}$$

$$= 0 \tag{59}$$

$$\sum_{i=1}^{d_x} \frac{\partial D}{\partial (\sigma_G)_i} \frac{\partial (\sigma'_G)_i}{\partial z_j} \frac{\partial G_i}{\partial z_k} = \sum_{i=1}^{d_x} \frac{\partial D}{\partial (\sigma_G)_i} \frac{\partial (\sigma'_G)_i}{\partial G_i} \frac{\partial G_i}{\partial z_j} \frac{\partial G_i}{\partial z_k} \tag{60}$$

$$= \sum_{i=1}^{d_x} \frac{\partial D}{\partial (\sigma_G)_i} \left( 2(\sigma_G)_i^3 - 2(\sigma_G)_i \right) \frac{\partial G_i}{\partial z_j} \frac{\partial G_i}{\partial z_k} \tag{61}$$

$$\sum_{i=1}^{d_x} \frac{\partial D}{\partial (\sigma_G)_i} (\sigma'_G)_i \frac{\cancel{\partial^2 G_i}}{\partial z_j \partial z_k} = 0 \tag{62}$$

Therefore, the simplified element is

$$(\nabla^2_{\mathbf{z}} \ell(\mathbf{z}))_{jk} = -\sum_{i=1}^{d_x} \frac{\partial D}{\partial (\sigma_G)_i} \left( 2(\sigma_G)_i^3 - 2(\sigma_G)_i \right) \frac{\partial G_i}{\partial z_j} \frac{\partial G_i}{\partial z_k} \tag{63}$$

We now vectorize the expression of $\nabla^2_{\mathbf{z}} \ell(\mathbf{z})$ as follows:

$$\nabla^2_{\mathbf{z}} \ell(\mathbf{z}) = -2 \left( \frac{\partial G}{\partial \mathbf{z}} \right)^T \operatorname{diag} \left( \frac{\partial D}{\partial \boldsymbol{\sigma}_G} \right) \operatorname{diag} \left( \boldsymbol{\sigma}_G^{\circ 3} - \boldsymbol{\sigma}_G \right) \left( \frac{\partial G}{\partial \mathbf{z}} \right) \tag{64}$$

Plugging this to Equation 47, the following is obtained:

$$\nabla_{\Sigma_c} \mathcal{L} = -\mathbb{E}_{q(\mathbf{z})} \left[ \pi_c \delta(\mathbf{z})_c \nabla_{\mathbf{z}}^T G \operatorname{diag} \left( \nabla_{\boldsymbol{\sigma}_G} D \odot (\boldsymbol{\sigma}_G^{\circ 3} - \boldsymbol{\sigma}_G) \right) \nabla_{\mathbf{z}} G \right] \tag{65}$$

$\square$

# D    METHODOLOGICAL DETAILS

## D.1    ADDITIVE ANGULAR MARGIN

To enhance the discriminative power of U2C loss, we adopted the additive angular margin (Deng et al., 2019) as follows:

$$\ell_{\text{U2C}}(\mathbf{z}^i) = - \log \frac{\exp(s \cdot \cos(\theta_{ii} + m))}{\frac{1}{B}\{\exp(s \cdot \cos(\theta_{ii} + m)) + \sum_{j \neq i} \exp(s \cdot \cos \theta_{ij})\}} \tag{66}$$

where $s$ denotes the feature scale, and $m$ is the angular margin. The feature scale $m$ and the coefficient of U2C loss $\lambda$ are linearly decayed to 0 during training, so that SLOGAN can focus more on the adversarial loss as training progresses.

## D.2    CLUSTER ASSIGNMENT

In Section 3.3, we chose $\cos \theta_{ic}$ as the critic function assuming that it is proportional to $\log p(c|\mathbf{x}_g)$. If the real data distribution $p(\mathbf{x}_r)$ and the generator distribution $p(\mathbf{x}_g)$ are sufficiently similar via adversarial learning, the cosine similarity between $E(\mathbf{x}_r)$ and $\boldsymbol{\mu}_c$ can also be considered proportional to $\log p(c|\mathbf{x}_r)$. Therefore, for real data, we obtain the probability for each cluster as follows:

$$\hat{p}(c|\mathbf{x}_r) = \frac{\exp(\cos \theta_c)}{\sum_{k=1}^{K} \exp(\cos \theta_k)} \tag{67}$$

where $\cos \theta_k = E(\mathbf{x}_r) \cdot \boldsymbol{\mu}_k / \|E(\mathbf{x}_r)\|\|\boldsymbol{\mu}_k\|$ is the cosine similarity between $E(\mathbf{x}_r)$ and $\boldsymbol{\mu}_k$. The data can then be assigned to the cluster with the highest probability (i.e., $\operatorname{argmax}_c \hat{p}(c|\mathbf{x}_r)$).

## D.3    ATTRIBUTE MANIPULATION

We utilized mixup (Zhang et al., 2018) to make the best use of a small amount of probe data when manipulating attributes. Algorithm 3 describes the procedure for using mixup for attribute manipulation when $K = 2$. We applied the same feature scale and angular margin to $\mathcal{L}_{\text{m}}$, as shown in Equation 66. The number of iterations for the mixup ($T$) was set to five.

---

**Algorithm 3** Attribute manipulation

---

Initialize probe data with the desired attribute $\mathbf{X}_{c=1} \leftarrow \{\mathbf{x}_{c=1}^i\}_{i=1}^M$ and $\bar{\mathbf{X}}_{c=1} \leftarrow \{\mathbf{x}_{c=1}^i\}_{i=1}^M$
Initialize probe data without the desired attribute $\mathbf{X}_{c=0} \leftarrow \{\mathbf{x}_{c=0}^i\}_{i=1}^M$ and $\bar{\mathbf{X}}_{c=0} \leftarrow \{\mathbf{x}_{c=0}^i\}_{i=1}^M$
**for** each mixup iteration $t$ in $\{1, ..., T\}$ **do**
    $\bar{\mathbf{X}}_{c=1} \leftarrow \bar{\mathbf{X}}_{c=1} \cup \text{MIXUP}(\mathbf{X}_{c=1}, \text{PERMUTE}(\mathbf{X}_{c=1}))$
    $\bar{\mathbf{X}}_{c=0} \leftarrow \bar{\mathbf{X}}_{c=0} \cup \text{MIXUP}(\mathbf{X}_{c=0}, \text{PERMUTE}(\mathbf{X}_{c=0}))$
**end for**
**for** each augmented data index $j$ in $\{1, ..., M(T+1)\}$ **do**
    $\cos \theta_{00}^j \leftarrow E(\bar{\mathbf{x}}_{c=0}^j) \cdot \boldsymbol{\mu}_0 / \|E(\bar{\mathbf{x}}_{c=0}^j)\|\|\boldsymbol{\mu}_0\|$
    $\cos \theta_{01}^j \leftarrow E(\bar{\mathbf{x}}_{c=0}^j) \cdot \boldsymbol{\mu}_1 / \|E(\bar{\mathbf{x}}_{c=0}^j)\|\|\boldsymbol{\mu}_1\|$
    $\cos \theta_{10}^j \leftarrow E(\bar{\mathbf{x}}_{c=1}^j) \cdot \boldsymbol{\mu}_0 / \|E(\bar{\mathbf{x}}_{c=1}^j)\|\|\boldsymbol{\mu}_0\|$
    $\cos \theta_{11}^j \leftarrow E(\bar{\mathbf{x}}_{c=1}^j) \cdot \boldsymbol{\mu}_1 / \|E(\bar{\mathbf{x}}_{c=1}^j)\|\|\boldsymbol{\mu}_1\|$
**end for**
$\mathcal{L}_{\text{m}} = -\frac{1}{M(T+1)} \sum_{j=1}^{M(T+1)} \log \frac{\exp(s \cdot \cos(\theta_{00}^j + m)) + \exp(s \cdot \cos(\theta_{11}^j + m))}{\exp(s \cdot \cos(\theta_{00}^j + m)) + \exp(s \cdot \cos \theta_{01}^j) + \exp(s \cdot \cos \theta_{10}^j) + \exp(s \cdot \cos(\theta_{11}^j + m))}$
Minimize $\mathcal{L}^p$ with respect to $E$, $G$, and $\boldsymbol{\mu}$

---

## D.4    SIMCLR

For the CIFAR and CelebA datasets, we used the SimCLR loss (Chen et al., 2020) for the encoder. We applied color, translation, and cutout transformations to the generated data using DiffAugment[1]

---

[1] https://github.com/mit-han-lab/data-efficient-gans

(Zhao et al., 2020). The SimCLR loss is calculated using the generated data $\mathbf{x}_g^i$ and augmentation $A$ as follows:

$$\ell_{\text{SimCLR}}(\mathbf{z}^i) = -\log \frac{\exp(E(\mathbf{x}_g^i) \cdot E(A(\mathbf{x}_g^i)) \ / \ \|E(\mathbf{x}_g^i)\| \|E(A(\mathbf{x}_g^i))\|)}{\sum_{j=1}^B \exp(E(\mathbf{x}_g^i) \cdot E(A(\mathbf{x}_g^j)) \ / \ \|E(\mathbf{x}_g^i)\| \|E(A(\mathbf{x}_g^j))\|)} \tag{68}$$

where $\mathbf{x}_g^i = G(\mathbf{z}^i)$. The encoder $E$ is trained to minimize $\frac{1}{B}\sum_{i=1}^B \left(\ell_{\text{adv}}(\mathbf{z}^i) + \ell_{\text{SimCLR}}(\mathbf{z}^i) + \lambda\ell_{\text{U2C}}(\mathbf{z}^i)\right)$.

## D.5 DeLiGAN+

Among the existing unsupervised conditional GANs, DeLiGAN lacks an encoder network. Therefore, for a fair comparison, we added an encoder network and named it DeLiGAN+. We set the output dimension of the encoder to equal the number of mixture components of the latent distribution. For the $i$-th example in the batch, when the $c_i$-th mixture component of the latent distribution is selected, DeLiGAN+ is learned through the following objective:

$$\min_{G,E,\mu_c,\sigma_c} \max_D \frac{1}{B}\sum_{i=1}^B \left[ D(\mathbf{x}^i) - D(G(\mathbf{z}^i, \mathbf{c}_i)) - \lambda_{CE}\, \mathbf{c}_i^T \log E(G(\mathbf{z}^i, \mathbf{c}_i)) \right] \tag{69}$$

where $\mathbf{c}_i$ is the one-hot vector corresponding to $c_i$ and $\lambda_{CE}$ is the coefficient of the cross entropy loss. We set $\lambda_{CE}$ to 10 in the experiments.

## D.6 Evaluation Metric

**Cluster assignment** We do not use clustering purity which is an evaluation metric for cluster assignment. To compute the clustering purity, the most frequent class in the cluster is obtained, and the ratio of the data points belonging to the class is calculated. However, if the attributes in the data are imbalanced, multiple clusters can be assigned to a single class in duplicate, and this high clustering purity misleads the results. Therefore, we utilized the normalized mutual information (NMI) implemented in scikit-learn[2].

**Unconditional generation** FID has the advantage of considering not only sample quality but also diversity, whereas Inception score (IS) cannot assess the diversity properly because IS does not compare generated samples with real samples (Shmelkov et al., 2018). Therefore, we used FID as the evaluation metric for unsupervised generation.

**Unsupervised conditional generation** If attributes in data are severely imbalanced, FID does not increase (deteriorate) considerably even if the model does not generate data containing the minority attributes. Therefore, the FID cannot accurately measure the unsupervised conditional generation performance for data with severely imbalanced attributes. We introduce ICFID to evaluate the performance of unsupervised conditional generation. When calculating ICFID, multiple clusters cannot be assigned to a single class in duplicate. Therefore, if data of a single class are generated from multiple discrete latent variables or modes, the model shows high (bad) ICFID.

---

[2]https://github.com/scikit-learn/scikit-learn/blob/15a949460/sklearn/
metrics/cluster/_supervised.py

# E    IMPLEMENTATION DETAILS

## E.1    GENERAL SETTINGS AND ENVIRONMENTS

For simplicity, we denote the learning rate of G as $\eta$ and the learning rate of $\boldsymbol{\Sigma}$ as $\gamma$. Throughout the experiments, we set the learning rate of $E$ to $\eta$, and $D$ to $4\eta$ using the two-timescale update rule (TTUR) (Heusel et al., 2017). We set the learning rate of $\boldsymbol{\mu}$ to $10\gamma$, and the learning rate of $\boldsymbol{\rho}$ to $\gamma$. We set $B$ to 64 and the number of training steps to 100k. To stabilize discriminator learning, we used Lipschitz penalty (Petzka et al., 2018) for the synthetic, MNIST, FMNIST, and 10x_73k datasets, and adversarial Lipschitz regularization (Terjék, 2020) for the CIFAR-10 and CelebA datasets. We repeated each experiment 3 times and reported the means and standard deviations of model performances. Hyperparameters are determined by a grid search. We used the Adam optimizer (Kingma & Ba, 2014) for training $D$, $G$, and $E$, and a gradient descent optimizer for training $\boldsymbol{\mu}$, $\boldsymbol{\Sigma}$, and $\boldsymbol{\rho}$. The experiments herein were conducted on a machine equipped with an Intel Xeon Gold 6242 CPU and an NVIDIA Quadro RTX 8000 GPU. The code is implemented in Python 3.7 and Tensorflow 1.14 (Abadi et al., 2016).

## E.2    SYNTHETIC DATASET

For the synthetic dataset, we first set the mean of eight 2-dimensional Gaussian distributions as $(0, 2)$, $(\sqrt{2}, \sqrt{2})$, $(2, 0)$, $(\sqrt{2}, -\sqrt{2})$, $(0, -2)$, $(-\sqrt{2}, -\sqrt{2})$, $(-2, 0)$, and $(-\sqrt{2}, \sqrt{2})$, and the variance as $0.01I$. In Figure 11, we also set the variances as $0.05I$ and $0.1I$. The number of data sampled from the Gaussian distributions was set to 5,000, 5,000, 5,000, 5,000, 15,000, 15,000, 15,000, and 15,000. We scaled a total of 80,000 data points to a range between -1 and 1. Table 11 shows the network architectures of SLOGAN used for the synthetic dataset. Linear $n$ denotes a fully-connected layer with $n$ output units. BN and SN denote batch normalization and spectral normalization, respectively. LReLU denotes the leaky ReLU. We set $\lambda = 4$, $\eta = 0.001$, $\gamma = 0.01$, $s = 2$, and $m = 0.5$.

Table 11: SLOGAN architecture used for the synthetic dataset

| $G$ | $D$ | $E$ |
|---|---|---|
| $\mathbf{z} \in \mathbb{R}^{64}$ | $\mathbf{x} \in \mathbb{R}^{2}$ | $\mathbf{x} \in \mathbb{R}^{2}$ |
| Linear 128 + BN + ReLU | Linear 128 + LReLU | Linear 128 + SN + LReLU |
| Linear 128 + BN + ReLU | Linear 128 + LReLU | Linear 128 + SN + LReLU |
| Linear 2 + Tanh | Linear 1 | Linear 64 + SN |

## E.3    MNIST AND FASHION-MNIST DATASETS

The MNIST dataset (LeCun et al., 1998) consists of handwritten digits, and the Fashion-MNIST (FMNIST) dataset (Xiao et al., 2017) is comprised of fashion products. Both the MNIST and Fashion-MNIST datasets have 60,000 training and 10,000 test 28×28 grayscale images. Each pixel was scaled to a range of $0-1$. The datasets consist of 10 classes, and the number of data points per class is balanced. Table 12 shows the network architectures of SLOGAN used for the MNIST and FMNIST datasets. Conv $k \times k$, $s$, $n$ denotes a convolutional network with $n$ feature maps, filter size $k \times k$, and stride $s$. Deconv $k \times k$, $s$, $n$ denotes a deconvolutional network with $n$ feature maps, filter size $k \times k$, and stride $s$. For the MNIST dataset, we set $\lambda = 10$, $\eta = 0.0001$, $\gamma = 0.002$, $s = 8$, and $m = 0.5$. For MNIST-2, we set $\lambda = 4$, $\eta = 0.0001$, $\gamma = 0.002$, $s = 4$, and $m = 0.5$. For the FMNIST dataset, we set $\lambda = 10$, $\eta = 0.0001$, $\gamma = 0.001$, $s = 1$, and $m = 0$. For FMNIST-5, we set $\lambda = 1$, $\eta = 0.0002$, $\gamma = 0.004$, $s = 4$, and $m = 0.5$.

## E.4    10X_73K DATASET

The 10x_73k dataset (Zheng et al., 2017) consists of 73,233 720-dimensional vectors, which are obtained from RNA transcript counts, and has eight cell types (classes). The number of data points per cell type is 10,085, 2,612, 9,232, 8,385, 10,224, 11,953, 10,479, and 10,263. We converted each element to logscale (i.e., $\log_2(x + 1)$) and scaled each element to a range between 0 and 1. Table 13 shows the network architectures of SLOGAN used for the 10x_73k dataset. We set $\lambda = 10$, $\eta = 0.0001$, $\gamma = 0.004$, $s = 4$, and $m = 0$.

Table 12: SLOGAN architecture used for the MNIST and FMNIST datasets

| G | D | E |
|---|---|---|
| $\mathbf{z} \in \mathbb{R}^{1 \times 1 \times 64}$ | $\mathbf{x} \in \mathbb{R}^{28 \times 28 \times 1}$ | $\mathbf{x} \in \mathbb{R}^{28 \times 28 \times 1}$ |
| Deconv 1×1, 1, 1024 + BN + ReLU | Conv 4×4, 2, 64 + LReLU | Conv 4×4, 2, 64 + LReLU |
| Deconv 7×7, 1, 128 + BN + ReLU | Conv 4×4, 2, 64 + LReLU | Conv 4×4, 2, 64 + LReLU |
| Deconv 4×4, 2, 64 + BN + ReLU | Conv 7×7, 1, 1024 + LReLU | Conv 7×7, 1, 1024 + LReLU |
| Deconv 4×4, 2, 1 + Sigmoid | Conv 1×1, 1, 1 | Conv 1×1, 1, 64 |

Table 13: SLOGAN architecture used for the 10x_73k dataset

| G | D | E |
|---|---|---|
| $\mathbf{z} \in \mathbb{R}^{64}$ | $\mathbf{x} \in \mathbb{R}^{2}$ | $\mathbf{x} \in \mathbb{R}^{2}$ |
| Linear 256 + LReLU | Linear 256 + LReLU | Linear 256 + SN + LReLU |
| Linear 256 + LReLU | Linear 256 + LReLU | Linear 256 + SN + LReLU |
| Linear 720 | Linear 1 | Linear 64 + SN |

## E.5 CIFAR-10 DATASET

The CIFAR-10 (Krizhevsky et al., 2009) dataset comprises 50,000 training and 10,000 test 32×32 color images. Each pixel was scaled to a range of -1 to 1. The number of data points per class is balanced. Figure 23 and Table 14 show the network architectures of residual blocks and SLOGAN used for the CIFAR datasets. AvgPool and GlobalAvgPool denote the average pooling and global average pooling layers, respectively. For the CIFAR-10, CIFAR-2, CIFAR-2 (7:3), and CIFAR-2 (9:1) datasets, we set $\lambda = 1$, $\eta = 0.0001$, $\gamma = 0.002$, $s = 4$, and $m = 0.5$.

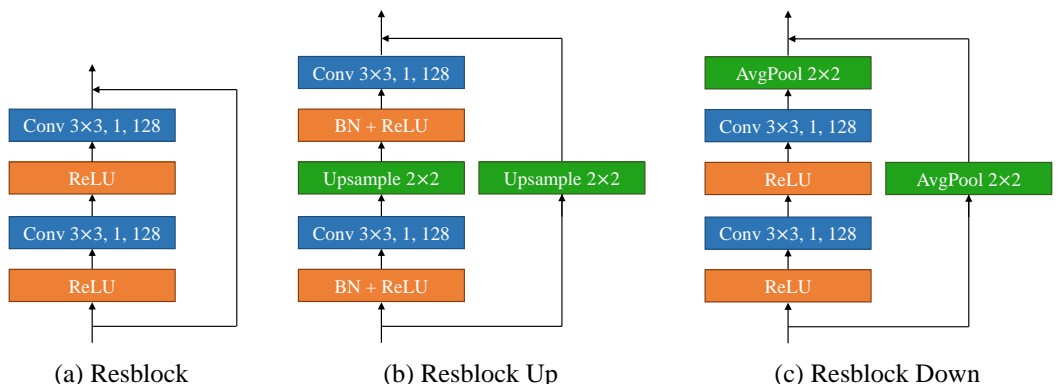

(a) Resblock        (b) Resblock Up        (c) Resblock Down

Figure 23: Resblock architectures used for colored image datasets.

Table 14: SLOGAN architecture used for the CIFAR datasets

| G | D | E |
|---|---|---|
| $\mathbf{z} \in \mathbb{R}^{128}$ | $\mathbf{x} \in \mathbb{R}^{32 \times 32 \times 3}$ | $\mathbf{x} \in \mathbb{R}^{32 \times 32 \times 3}$ |
| Linear 2048 + Reshape 4, 4, 128 | Resblock Down | Resblock Down |
| Resblock Up | Resblock Down | Resblock Down |
| Resblock Up | Resblock | Resblock |
| Resblock Up | Resblock | Resblock |
| BN + ReLU | ReLU + GlobalAvgPool | ReLU + GlobalAvgPool |
| Conv 3×3, 1, 3 + Tanh | Linear 1 | Linear 128 |

## E.6 CELEBA DATASET

The CelebA dataset (Liu et al., 2015) consists of 202,599 face attributes. We cropped the face part of each image to 140×140 pixels, resized it to 64×64 pixels, and scaled it to a range between -1 and 1.

The imbalanced ratio is different for each attribute, and the imbalanced ratios of male and eyeglasses used in the experiment are 1:1 and 14:1, respectively. Table 15 lists the network architectures of SLOGAN used for the CelebA dataset. We set $\lambda = 1$, $\eta = 0.0002$, $\gamma = 0.0006$, $s = 4$, and $m = 0.5$.

Table 15: SLOGAN architecture used for the CelebA dataset

| $G$ | $D$ | $E$ |
|---|---|---|
| $\mathbf{z} \in \mathbb{R}^{128}$ | $\mathbf{x} \in \mathbb{R}^{64 \times 64 \times 3}$ | $\mathbf{x} \in \mathbb{R}^{64 \times 64 \times 3}$ |
| Linear 8192 + Reshape 8, 8, 128 | Resblock Down | Resblock Down |
| Resblock Up | Resblock Down | Resblock Down |
| Resblock Up | Resblock Down | Resblock Down |
| Resblock Up | Resblock | Resblock |
| BN + ReLU | ReLU + GlobalAvgPool | ReLU + GlobalAvgPool |
| Conv 3×3, 1, 3 + Tanh | Linear 1 | Linear 128 |

### E.7 CELEBA-HQ DATASET

The CelebA-HQ dataset (Karras et al., 2017) consists of 30,000 face attributes. We resized each image to 128×128 and 256×256 pixels, and scaled it to a range between -1 and 1. The imbalance ratio of male used in the experiment was 1.7:1. We used StyleGAN2 (Karras et al., 2020) architecture with DiffAugment[3] and applied implicit reparameterization to the input space of the mapping network. We set $\lambda = 1$, $\eta = 0.002$, $\gamma = 0.0006$, $s = 1$, and $m = 0$.

### E.8 AFHQ DATASET

The AFHQ dataset (Choi et al., 2020) consists of 15,000 high-quality animal faces. We used cats and dogs, and there are about 5,000 images each in the dataset. We resized each image to 256×256 pixels, and scaled it to a range between -1 and 1. We set the imbalance ratios of cats and dogs to 1:1, 1:2, and 1:5 by reducing the number of cats in the training dataset. We used the same model architecture and hyperparameters as for the CelebA-HQ dataset.

### E.9 CODE AVAILABILITY

Code is available at `https://github.com/shinyflight/SLOGAN`

---

[3]`https://github.com/mit-han-lab/data-efficient-gans/tree/master/`
`DiffAugment-stylegan2`

## REFERENCES IN APPENDIX

Martín Abadi, Paul Barham, Jianmin Chen, Zhifeng Chen, Andy Davis, Jeffrey Dean, Matthieu Devin, Sanjay Ghemawat, Geoffrey Irving, Michael Isard, et al. Tensorflow: A system for large-scale machine learning. In *12th USENIX symposium on operating systems design and implementation (OSDI 16)*, pp. 265–283, 2016.

Martin Arjovsky, Soumith Chintala, and Léon Bottou. Wasserstein generative adversarial networks. In *International conference on machine learning*, pp. 214–223. PMLR, 2017.

Mohammadreza Armandpour, Ali Sadeghian, Chunyuan Li, and Mingyuan Zhou. Partition-guided gans. In *Proceedings of the IEEE/CVF Conference on Computer Vision and Pattern Recognition*, pp. 5099–5109, 2021.

Mohamed Ishmael Belghazi, Sai Rajeswar, Olivier Mastropietro, Negar Rostamzadeh, Jovana Mitrovic, and Aaron Courville. Hierarchical adversarially learned inference. *arXiv preprint arXiv:1802.01071*, 2018.

Yoshua Bengio, Aaron Courville, and Pascal Vincent. Representation learning: A review and new perspectives. *IEEE transactions on pattern analysis and machine intelligence*, 35(8):1798–1828, 2013.

Ting Chen, Simon Kornblith, Mohammad Norouzi, and Geoffrey Hinton. A simple framework for contrastive learning of visual representations. In *International conference on machine learning*, pp. 1597–1607. PMLR, 2020.

Xi Chen, Yan Duan, Rein Houthooft, John Schulman, Ilya Sutskever, and Pieter Abbeel. Infogan: interpretable representation learning by information maximizing generative adversarial nets. In *Neural Information Processing Systems (NIPS)*, 2016.

Jacob Cohen. *Statistical power analysis for the behavioral sciences*. Routledge, 2013.

Jiankang Deng, Jia Guo, Niannan Xue, and Stefanos Zafeiriou. Arcface: Additive angular margin loss for deep face recognition. In *Proceedings of the IEEE/CVF Conference on Computer Vision and Pattern Recognition*, pp. 4690–4699, 2019.

Jeff Donahue, Philipp Krähenbühl, and Trevor Darrell. Adversarial feature learning. In *5th International Conference on Learning Representations*, 2017.

Vincent Dumoulin, Ishmael Belghazi, Ben Poole, Alex Lamb, Martín Arjovsky, Olivier Mastropietro, and Aaron C. Courville. Adversarially learned inference. In *5th International Conference on Learning Representations*, 2017.

Martin Heusel, Hubert Ramsauer, Thomas Unterthiner, Bernhard Nessler, and Sepp Hochreiter. Gans trained by a two time-scale update rule converge to a local nash equilibrium. In *Proceedings of the 31st International Conference on Neural Information Processing Systems*, pp. 6629–6640, 2017.

Ayush Jaiswal, Wael AbdAlmageed, Yue Wu, and Premkumar Natarajan. Bidirectional conditional generative adversarial networks. In *Asian Conference on Computer Vision*, pp. 216–232. Springer, 2018.

Minguk Kang and Jaesik Park. Contragan: Contrastive learning for conditional image generation. In *Advances in Neural Information Processing Systems*, 2020.

Diederik P Kingma and Jimmy Ba. Adam: A method for stochastic optimization. *arXiv preprint arXiv:1412.6980*, 2014.

Alex Krizhevsky, Geoffrey Hinton, et al. Learning multiple layers of features from tiny images. 2009.

Yann LeCun, Léon Bottou, Yoshua Bengio, and Patrick Haffner. Gradient-based learning applied to document recognition. *Proceedings of the IEEE*, 86(11):2278–2324, 1998.

Wu Lin, Mohammad Emtiyaz Khan, and Mark Schmidt. Stein's lemma for the reparameterization trick with exponential family mixtures. *arXiv preprint arXiv:1910.13398*, 2019.

Steven Liu, Tongzhou Wang, David Bau, Jun-Yan Zhu, and Antonio Torralba. Diverse image generation via self-conditioned gans. In *Proceedings of the IEEE/CVF Conference on Computer Vision and Pattern Recognition*, pp. 14286–14295, 2020.

Ziwei Liu, Ping Luo, Xiaogang Wang, and Xiaoou Tang. Deep learning face attributes in the wild. In *Proceedings of International Conference on Computer Vision (ICCV)*, December 2015.

Deepak Mishra, Aravind Jayendran, and AP Prathosh. Effect of the latent structure on clustering with gans. *IEEE Signal Processing Letters*, 27:900–904, 2020.

Takeru Miyato and Masanori Koyama. cgans with projection discriminator. In *International Conference on Learning Representations*, 2018.

Sudipto Mukherjee, Himanshu Asnani, Eugene Lin, and Sreeram Kannan. Clustergan: Latent space clustering in generative adversarial networks. In *Proceedings of the AAAI conference on artificial intelligence*, volume 33, pp. 4610–4617, 2019.

Augustus Odena, Christopher Olah, and Jonathon Shlens. Conditional image synthesis with auxiliary classifier gans. In *International conference on machine learning*, pp. 2642–2651. PMLR, 2017.

Lili Pan, Peijun Tang, Zhiyong Chen, and Zenglin Xu. Contrastive disentanglement in generative adversarial networks. *arXiv preprint arXiv:2103.03636*, 2021.

Henning Petzka, Asja Fischer, and Denis Lukovnikov. On the regularization of wasserstein GANs. In *International Conference on Learning Representations*, 2018.

Konstantin Shmelkov, Cordelia Schmid, and Karteek Alahari. How good is my gan? In *Proceedings of the European Conference on Computer Vision (ECCV)*, pp. 213–229, 2018.

Dávid Terjék. Adversarial lipschitz regularization. In *International Conference on Learning Representations*, 2020.

Wouter Van Gansbeke, Simon Vandenhende, Stamatios Georgoulis, Marc Proesmans, and Luc Van Gool. Scan: Learning to classify images without labels. In *European Conference on Computer Vision*, pp. 268–285. Springer, 2020.

Frank Wilcoxon. Individual comparisons by ranking methods. In *Breakthroughs in statistics*, pp. 196–202. Springer, 1992.

Han Xiao, Kashif Rasul, and Roland Vollgraf. Fashion-mnist: a novel image dataset for benchmarking machine learning algorithms. *arXiv preprint arXiv:1708.07747*, 2017.

Hongyi Zhang, Moustapha Cisse, Yann N. Dauphin, and David Lopez-Paz. mixup: Beyond empirical risk minimization. In *International Conference on Learning Representations*, 2018.

Shengyu Zhao, Zhijian Liu, Ji Lin, Jun-Yan Zhu, and Song Han. Differentiable augmentation for data-efficient gan training. *Advances in Neural Information Processing Systems*, 33, 2020.

Grace XY Zheng, Jessica M Terry, Phillip Belgrader, Paul Ryvkin, Zachary W Bent, Ryan Wilson, Solongo B Ziraldo, Tobias D Wheeler, Geoff P McDermott, Junjie Zhu, et al. Massively parallel digital transcriptional profiling of single cells. *Nature communications*, 8(1):1–12, 2017.

