# OpenReview forum: "Stein Latent Optimization for Generative Adversarial Networks"
_ICLR.cc/2022/Conference — ICLR 2022 Poster_

### Official Review · Reviewer_9K6Q · 2021-11-03

**Correctness:** 4
**Technical Novelty And Significance:** 3
**Empirical Novelty And Significance:** 3
**Recommendation:** 6
**Confidence:** 3

**Main Review:**

I found this approach to be quite reasonable, and it consists of components that work well together. While GANs with mixture latent spaces have been proposed before, I am not aware of a method that efficiently updates the prior components. Previous work has used the reparameterization trick to derive component-specific updates, so the novelty of this specific component in using implicit reparameterization is a bit low, but the benefit is still there. I thought the introduction of the contrastive loss was interesting, but I had some concerns about the diversity of generated samples when the encodings are encouraged to match just the mean vector, ignoring the covariances. In the experiments, it looks to perform quite well, but I wonder how this would perform with less tightly-clustered data (even 8 Gaussians but with larger covariances).

From the experiments, I mostly wanted to see how this method handles a large imbalance on high-dimensional data. I am somewhat satisfied with the CelebA-HQ results with a ratio of 1.7:1, but this seems a bit low and does not stress the model at all. I wonder where does the model start to struggle. Is 2:1 possible?  Or even 5:1? Figure 5 examines this a little, but eyeglass is an attribute with global features almost identical to no-eyeglasses, so I don't think it is very informative.

-----------------------------------------
### Post Rebuttal Comments

I thank the authors for their additional efforts in addressing my concerns. I am satisfied with this response and I will be retaining my score.


**Summary Of The Paper:**

This work proposes a method for performing unsupervised conditional generation from imbalanced attributes. The key components of this work are the following:

* A (architecture agnostic) GAN with a GMM latent space, where we have one component per attribute in the data.

* The parameters of the components $(\mu, \Sigma)$ are learned via implicit reparameterization, and leveraging stein's lemma to derive gradients for $\mu$ and $\Sigma$.

* The components are encouraged to be distinct by introducing a contrastive objective. An encoder is used to obtain a low dimensional vector representation of a generated sample, and this sample is encouraged to be close to the mean vector of the corresponding component in the latent space.

**Summary Of The Review:**

I think this is a reasonable combination of components that make progress on the task: facilitating learning of samplers from data with highly imbalanced class attributes. Given the performance increase compared to [1], it looks like using implicit reparameterization and a contrastive loss (the main novelty here) make a clear difference in performance. I think each of the components are justified and perform well.

[1] Swaminathan Gurumurthy, Ravi Kiran Sarvadevabhatla, and R Venkatesh Babu. Deligan: Generative adversarial networks for diverse and limited data. In Proceedings of the IEEE conference on computer vision and pattern recognition, pp. 166–174, 2017.

---

> ### Author Response · Authors · 2021-11-19
> **Response to Reviewer 9K6Q**
>
> We thank you for the positive comments and suggestions. We have addressed each of your questions below.
>
> **Q1.** I thought the introduction of the contrastive loss was interesting, but I had some concerns about the diversity of generated samples when the encodings are encouraged to match just the mean vector, ignoring the covariances. In the experiments, it looks to perform quite well, but I wonder how this would perform with less tightly-clustered data (even 8 Gaussians but with larger covariances).
>
> **Response:** Thank you for the good question. For U2C loss, we find the mean vector corresponding to the component ID with the highest responsibility $\arg\max_c q(c|\mathbf{z})$. Since the responsibility of a component is calculated using both the mean vector and covariance, covariances are also considered for the contrastive loss.
>
> As discussed in Self-conditioned GAN [1] and PGMGAN [2] papers, partitioning of the data space improves the diversity of generated samples. From this point of view, U2C loss plays role in partitioning the data space. In our method described in Appendix D.1, the feature scale and the coefficient of the contrastive loss are linearly decayed to 0 during training. At the beginning of the training, SLOGAN focuses more on breaking the data space into smaller regions. As training progresses, SLOGAN focuses on generating realistic and diverse samples in the small regions of the data space. In this way, U2C loss rather improves the diversity of generated samples.
>
> Following your suggestion, we conducted new experiments on the synthetic datasets with various covariances. As shown in Figure 11 in the revision, SLOGAN successfully performs unsupervised conditional generation on the synthetic datasets with larger covariances ($\sigma=0.05$ and $\sigma=0.10$).
>
> **Q2.** From the experiments, I mostly wanted to see how this method handles a large imbalance on high-dimensional data. I am somewhat satisfied with the CelebA-HQ results with a ratio of 1.7:1, but this seems a bit low and does not stress the model at all. I wonder where does the model start to struggle. Is 2:1 possible? Or even 5:1? Figure 5 examines this a little, but eyeglass is an attribute with global features almost identical to no-eyeglasses, so I don't think it is very informative.
>
> **Response:** Thanks for the suggestion! We conducted new experiments on the AFHQ dataset with various imbalance ratios of cats and dogs. We qualitatively analyze SLOGAN without using probe data. We set the imbalance ratios of cats and dogs to 1:1, 1:2, and 1:5 by reducing the number of cats in the training dataset. When the imbalance ratios are 1:1 and 1:2, SLOGAN identifies cat/dog as the most salient attribute and learned the attribute successfully as presented in Figure 22 (a) and (b) in the revision. However, as shown in Figure 22 (c) in the revision, SLOGAN discovers folded ears as the most salient attribute when the imbalance ratio is 1:5.
>
> We wish that our response has addressed your concerns. If you have any questions, please feel free to let us know during the rebuttal window. Thank you very much!
>
> [1] Diverse image generation via self-conditioned gans, CVPR 2020.\
> [2] Partition-Guided GANs, CVPR 2021.

---

> > ### Author Response · Authors · 2021-11-27
> > **Looking forward to your post-rebuttal feedback!**
> >
> > Dear Reviewer 9K6Q
> >
> > Thank you again for the insightful comments and suggestions! Since there are only very few days left, we sincerely look forward to your follow-up response. We are happy to provide additional answers to illustrate the strength of our approach.
> >
> > In our previous response, we have carefully read your comments and made detailed responses summarized below:
> >
> > - Provided additional explanation on how our method takes the covariances into account and how our method generates diverse samples.
> > - Conducted the new experiment to show SLOGAN can generate diverse samples.
> > - Conducted the new experiment to show our method identifies salient attributes on high-dimensional datasets with various imbalance ratios.
> >
> > We hope that the provided new experiments and the additional explanation have convinced you of the merits of this paper. If there are additional questions, please feel free to let us know.
> >
> > Meanwhile, we would like to thank the reviewer again for the very helpful comments. By taking them into account, it would indeed make our paper clearer and stronger.
> >
> > Thank you for your time and effort!
> >
> > Best regards, Authors

---

> > > ### Comment · Reviewer_9K6Q · 2021-11-29
> > > **Thanks for the comments**
> > >
> > > Sorry for the late response. I do appreciate the additional experiments and explanations.
> > > I have looked over the changes made to the manuscript and I now have a good grasp on how well this method handles increased imbalances. The results look consistent, with low dimensional experimental results matching those of high dimension.
> > > Due to the extensive experiments conducted and their consistent results, I will opt to keep my score the same

---

> > > > ### Author Response · Authors · 2021-11-30
> > > > **Thank you!**
> > > >
> > > > Thank you very much for your response. We are glad to know that your concerns have been addressed.

---

### Official Review · Reviewer_5GSd · 2021-11-03

**Correctness:** 3
**Technical Novelty And Significance:** 4
**Empirical Novelty And Significance:** 3
**Recommendation:** 6
**Confidence:** 4

**Details Of Ethics Concerns:**

None.

**Main Review:**

Strengths:
1. The idea of using Gaussian mixture as a prior for imbalanced attributes is intuitive and technically sound.
2. In the theoretical aspect, this paper derives Stein latent optimization based on Stein’s lemma to estimate the gradient of parameters.
3. The paper is well-written, and the result looks promising.

Weaknesses:
1. The imbalance datasets (MNIST and CIFAR) contain two classes. I would expect to evaluate over more imbalance settings of the datasets, such as one class is 0.1 fraction of the other nine classes.
2. Two important baselines in GANs latent space clustering are missing. “Effect of The Latent Structure on Clustering with GANs” and “Mixture of GANs for Clustering”.
3. Compared with ClusterGAN, this paper address the importance of generated image quality. The study may provide a comparison to Self-conditioned GAN or IC-GAN in terms of both generated image quality and clustering quality.
4. In the current version, it is difficult to infer how U2C loss helps the performance of the proposed model. I would expect the study presents the quantitative results (e.g., Tables 1 and 2) of the proposed without U2C loss.

*****************************

Post Rebuttal Comments:
The authors have addressed my concerns in the revised version. I am satisfied with the response and change my rating from 5 to 6.


**Summary Of The Paper:**

This paper uses Gaussian mixture as a prior to address the scenario of imbalanced attributes in GANs latent space clustering. The study derives Stein latent optimization that provides reparameterizable gradient estimations when assuming a Gaussian mixture prior for the latent space. The results show that the proposed model achieves superior performance compared to the baselines.

**Summary Of The Review:**

Although the derived Stein latent optimization for imbalanced attributes is interesting and promising, the current experiments are not enough to support the claims.

---

> ### Author Response · Authors · 2021-11-19
> **Response to Reviewer 5GSd (1/2)**
>
> We thank you for the insightful comments and suggestions. We have addressed each of your questions below.
>
> **Q1.** The imbalance datasets (MNIST and CIFAR) contain two classes. I would expect to evaluate over more imbalance settings of the datasets, such as one class is 0.1 fraction of the other nine classes.
>
> **Response:** Thanks for the suggestion! We conducted new experiments on highly imbalanced multi-class datasets by setting class 8 of the MNIST dataset to very low proportions of the other nine classes (e.g., 10:10:10:10:10:10:10:10:1:10 and 10:10:10:10:10:10:10:10:2:10). When class 8 is 0.1 fraction of the other nine classes, training images of class 7 with a horizontal line outnumber images of class 8, and SLOGAN identifies 7 with a horizontal line as a more salient attribute than 8 as shown in the red boxes in Figure 21 (a) in the revision. On the other hand, when class 8 is 0.2 fraction of the other nine classes, training images of class 8 outnumber images of class 7 with a horizontal line. SLOGAN successfully identifies 8 as a salient attribute as shown in the red box in Figure 21 (b) in the revision.
>
> **Q2.** Two important baselines in GANs latent space clustering are missing. “Effect of The Latent Structure on Clustering with GANs” and “Mixture of GANs for Clustering”.
>
> **Response:** NEMGAN [1] argues that it considers the imbalance problem of attributes. However, it assumes that some labeled samples are provided, which is unrealistic in real-world scenarios. In addition, labeled samples should have the same imbalance ratio as the training data to estimate the imbalance ratio. If these samples are not given, NEMGAN is the same method as ClusterGAN.
>
> We compared SLOGAN with NEMGAN and GANMM [2] on the MNIST dataset in terms of clustering metrics including purity, adjusted Rand index (ARI), and NMI. For a fair comparison, we did not provide labeled samples to NEMGAN, and the results are summarized in the following table:
>
> <Table S1.> Clustering performance comparison on the MNIST dataset.
>
> |Method|Purity$\uparrow$|ARI$\uparrow$|NMI$\uparrow$|
> |-----------|:-----------:|:-----------:|:-----------:|
> |NEMGAN|0.8698|0.7633|0.8058|
> |GANMM|0.8908|0.8361|0.8654|
> |**SLOGAN**|**0.9668**|**0.9283**|**0.9188**|
>
> From the result, we confirm that SLOGAN significantly outperforms NEMGAN and GANMM.

---

> ### Author Response · Authors · 2021-11-19
> **Response to Reviewer 5GSd (2/2)**
>
> **Q3.** Compared with ClusterGAN, this paper address the importance of generated image quality. The study may provide a comparison to Self-conditioned GAN or IC-GAN in terms of both generated image quality and clustering quality.
>
> **Response:** We appreciate your suggestion. Self-conditioned GAN is closely related to our work, but IC-GAN does not perform the same task as our work in that IC-GAN requires a data point to generate data. Instead, we additionally compared our method with PGMGAN [3] which is one of the state-of-the-art unsupervised conditional GANs. We conducted new comparative experiments with Self-conditioned GAN and PGMGAN and the results on balanced and imbalanced cifar datasets are summarized in the following tables.
>
> <Table S2.> Performance comparison on the CIFAR-2 dataset.
>
> |Method|NMI$\uparrow$|FID$\downarrow$|ICFID$\downarrow$|
> |-----------|:-----------:|:-----------:|:-----------:|
> |Self-conditioned GAN|0.00$\pm$0.00|39.44$\pm$1.72|71.54$\pm$5.41|
> |PGMGAN|0.67$\pm$0.00|29.49$\pm$0.51|**35.67$\pm$0.61**|
> |**SLOGAN**|**0.78$\pm$0.03**|**28.99$\pm$0.36**|35.68$\pm$0.51|
>
> <Table S3.> Performance comparison on the CIFAR-10 dataset.
>
> |Method|NMI$\uparrow$|FID$\downarrow$|ICFID$\downarrow$|
> |-----------|:-----------:|:-----------:|:-----------:|
> |Self-conditioned GAN|0.01$\pm$0.00|199.28$\pm$57.16|262.54$\pm$59.29|
> |PGMGAN|0.29$\pm$0.02|31.50$\pm$0.73|81.25$\pm$11.55|
> |**SLOGAN**|**0.34$\pm$0.01**|**20.61$\pm$0.40**|**71.23$\pm$6.76**|
>
> As shown in the results on the datasets with balanced attributes, SLOGAN outperforms Self-conditioned GAN and shows comparable performances with PGMGAN. Self-conditioned GAN has no loss to facilitate clustering of discriminator features, and the performance seems to be largely influenced by the architecture of the discriminator.
>
> <Table S4.> Performance comparison on the CIFAR-2 (7:3) dataset.
>
> |Method|NMI$\uparrow$|FID$\downarrow$|ICFID$\downarrow$|
> |-----------|:-----------:|:-----------:|:-----------:|
> |Self-conditioned GAN|0.00$\pm$0.00|45.28$\pm$1.81|88.58$\pm$4.57|
> |PGMGAN|0.42$\pm$0.03|29.76$\pm$1.65|57.06$\pm$3.31|
> |**SLOGAN**|**0.69$\pm$0.02**|**29.09$\pm$0.73**|**45.83$\pm$3.03**|
>
> <Table S5.> Performance comparison on the CIFAR-2 (9:1) dataset.
>
> |Method|NMI$\uparrow$|FID$\downarrow$|ICFID$\downarrow$|
> |-----------|:-----------:|:-----------:|:-----------:|
> |Self-conditioned GAN|0.00$\pm$0.00|50.45$\pm$1.56|123.35$\pm$6.56|
> |PGMGAN|0.16$\pm$0.03|30.23$\pm$1.31|101.68$\pm$3.87|
> |**SLOGAN**|**0.38$\pm$0.01**|**29.47$\pm$1.53**|**86.75$\pm$1.87**|
>
> As shown in the results on the datasets with imbalanced attributes, SLOGAN significantly outperforms the other methods. To avoid degenerated clusters where one partition contains most of the data, PGMGAN uses an entropy term that regularizes the average cluster probability vector to a uniform distribution. However, when learning PGMGAN on datasets with imbalanced attributes, it is difficult to adjust the coefficient of the regularizer because the performance seems to be sensitive to the strength of the regularizer in practice. We have added results on more datasets in Tables 1, 2, 6, and 7 in the revision. We also have cited these works to the background and discussed the above methods in Appendix B.2 in the revision.
>
> **Q4.** In the current version, it is difficult to infer how U2C loss helps the performance of the proposed model. I would expect the study presents the quantitative results (e.g., Tables 1 and 2) of the proposed without U2C loss.
>
> **Response:** Following your suggestion, we conducted additional ablation studies on U2C loss with various datasets and the results are summarized in the following table:
>
> <Table S6.> Ablation studies on U2C loss
>
> |Dataset|Ablation|NMI$\uparrow$|FID$\downarrow$|ICFID$\downarrow$|
> |-----------|-----------|:-----------:|:-----------:|:-----------:|
> |MNIST-2 (7:3)|SLOGAN w/o $\ell_\mathrm{U2C}$|0.25|4.62|9.43|
> ||SLOGAN|**0.92**|**4.02**|**5.91**|
> |FMNIST-5|SLOGAN w/o $\ell_\mathrm{U2C}$|0.14|**5.27**|43.15|
> ||SLOGAN|**0.66**|5.29|**32.46**|
> |CIFAR-2|SLOGAN w/o $\ell_\mathrm{U2C}$|0.01|29.18|41.72|
> ||SLOGAN|**0.78**|**28.99**|**35.68**|
> |CIFAR-2 (7:3)|SLOGAN w/o $\ell_\mathrm{U2C}$|0.08|30.34|48.82|
> ||SLOGAN|**0.69**|**29.09**|**45.83**|
> |CIFAR-10|SLOGAN w/o $\ell_\mathrm{U2C}$|0.08|20.91|78.26|
> ||SLOGAN|**0.34**|**20.61**|**71.23**|
>
> From the results, we confirm that U2C loss significantly improves the clustering and unsupervised conditional generation performances of SLOGAN. We have added the results in Table 9 in the revision.
>
> We wish that our response has addressed your concerns. If you have any questions, please feel free to let us know during the rebuttal window. Thank you very much!
>
> [1] Effect of The Latent Structure on Clustering with GANs, IEEE Signal Processing Letters 2020.\
> [2] Mixture of GANs for Clustering, IJCAI 2018.\
> [3] Partition-Guided GANs, CVPR 2021.

---

> > ### Author Response · Authors · 2021-11-27
> > **Looking forward to your post-rebuttal feedback!**
> >
> > Dear Reviewer 5GSd
> >
> > Thank you again for the insightful comments and suggestions! Since the deadline of discussion is approaching, we sincerely look forward to your follow-up response. We are happy to provide any additional clarification that you may need.
> >
> > In our previous response, we conducted additional experiments to support our claims. For your convenience, we provide a summary below:
> >
> > - Conducted the new experiments to show our method works robustly on highly imbalanced multi-class datasets.
> > - Provided experimental results to show that SLOGAN outperforms NEMGAN and GANMM in terms of clustering metrics.
> > - Provided additional discussion on why we did not compare our method with IC-GAN and suggested PGMGAN as an alternative state-of-the-art method for comparison.
> > - Conducted the new experiments to show that SLOGAN outperforms Self-conditioned GAN and PGMGAN.
> > - Conducted additional experiments to show the effectiveness of U2C loss.
> >
> > We hope that the provided new experiments and the additional discussion have convinced you of the merits of this paper. Please do not hesitate to contact us if there are additional questions.
> >
> > Meanwhile, we would like to thank the reviewer again for the very helpful comments. By taking them into account, it would indeed make our paper clearer and stronger.
> >
> > Thank you for your time and effort!
> >
> > Best regards, Authors

---

### Official Review · Reviewer_bra9 · 2021-11-03

**Correctness:** 4
**Technical Novelty And Significance:** 3
**Empirical Novelty And Significance:** 3
**Recommendation:** 6
**Confidence:** 3

**Main Review:**

Strength:
1. It is interesting to introduce Stein’s lemma to provide a first-order gradient identity for multivariate Gaussian distribution.
2. The way of using a reparameterization trick to estimate gradients of the parameters of Gaussian mixtures is quite neat.
3. The paper is well-written and clearly presented.

Weakness:
1. It is reasonable to conduct visualization comparisons with the most recent methods.
2. The authors could present failure cases that may be associated with different attributes.
3. How to obtain a suitable attribute ratio for the probe data? What will happen if we change the ratio from 14:1 to 1:14 for eyeglasses?

**Summary Of The Paper:**

The paper proposed a Stein Latent Optimization for Generative adversarial networks, which can perform conditional generation in an unsupervised manner. The core innovation is the reparameterizable gradient estimations. The proposed unsupervised conditional contrastive loss further ensures the single attribute generation capacity. The idea is somewhat novel and motivated. The authors perform empirical studies on diverse datasets to conclude that the proposed algorithm is effective in learning balanced or imbalanced attributes.

**Summary Of The Review:**

Overall, the paper is somewhat novel and easy to follow. The assumptions and decisions are well supported. The stepwise experiments are helpful and provide good insights to evaluate the proposed algorithm.

---

> ### Author Response · Authors · 2021-11-19
> **Response to Reviewer bra9**
>
> We thank you for the positive comments and suggestions. We have addressed each of your questions below.
>
> **Q1.** It is reasonable to conduct visualization comparisons with the most recent methods.
>
> **Response:** We conducted new visualization comparisons with the most recent methods such as CD-GAN (2021) [1] and PGMGAN (2021) [2] on the CIFAR-2 (7:3) dataset. From the results, we confirm that SLOGAN learns imbalanced attributes of the dataset most robustly. We have added the results in Figure 20 in the revision.
>
> **Q2.** The authors could present failure cases that may be associated with different attributes.
>
> **Response:** Thanks for the constructive suggestion! We newly conducted two experiments to present failure cases.
>
> First, we trained our method on highly imbalanced multi-class datasets by setting class 8 of the MNIST dataset to very low proportions of the other nine classes (e.g., 10:10:10:10:10:10:10:10:1:10 and 10:10:10:10:10:10:10:10:2:10). When class 8 is 0.1 fraction of the other nine classes, training images of class 7 with a horizontal line outnumber images of class 8, and SLOGAN identifies 7 with a horizontal line as a more salient attribute than 8 as shown in the red boxes in Figure 21 (a) in the revision. On the other hand, when class 8 is 0.2 fraction of the other nine classes, training images of class 8 outnumber images of class 7 with a horizontal line. SLOGAN successfully identifies 8 as a salient attribute as shown in the red box in Figure 21 (b) in the revision.
>
> Second, we trained SLOGAN on the AFHQ dataset with various imbalance ratios of cats and dogs. We qualitatively analyze SLOGAN without using probe data. We set the imbalance ratios of cats and dogs to 1:1, 1:2, and 1:5 by reducing the number of cats in the training dataset. When the imbalance ratios are 1:1 and 1:2, SLOGAN identifies cat/dog as the most salient attribute and learned the attribute successfully as presented in Figure 22 (a) and (b) in the revision. However, as shown in Figure 22 (c) in the revision, SLOGAN discovers folded ears as the most salient attribute when the imbalance ratio is 1:5.
>
> **Q3.** How to obtain a suitable attribute ratio for the probe data? What will happen if we change the ratio from 14:1 to 1:14 for eyeglasses?
>
> **Response:** The use of probe data does not depend on the imbalance ratio, but if multiple attributes co-exist in the training dataset, probe data need to be used to learn the desired attribute. For example, in the CelebA dataset, the number of male and female is balanced, but probe data are required to learn the Male attribute. On the other hand, in the CIFAR-2 (7:3) dataset, SLOGAN successfully learns the attributes without probe data, even though the dataset has imbalanced attributes. Regardless of the imbalance ratio for eyeglasses, it seems that probe data should be used because undesired attributes (e.g., background color, hair color, gender, …) may be learned.
>
> We wish that our response has addressed your concerns. If you have any questions, please feel free to let us know during the rebuttal window. Thank you very much!
>
> [1] Contrastive Disentanglement in Generative Adversarial Networks, arXiv 2021.\
> [2] Partition-Guided GANs, CVPR 2021.

---

> > ### Author Response · Authors · 2021-11-27
> > **Looking forward to your post-rebuttal feedback!**
> >
> > Dear Reviewer bra9
> >
> > Thank you again for the insightful comments and suggestions! Since there are only very few days left, we sincerely look forward to your follow-up response. We are happy to provide additional answers to illustrate the strength of our approach.
> >
> > In our previous response, we have carefully read your comments and made detailed responses summarized below:
> >
> > - Conducted the new experiment to show that our method qualitatively outperforms the most recent methods.
> > - Conducted the new experiments to provide failure cases that are associated with different attributes.
> > - Provided additional explanation on when probe data are needed.
> >
> > We hope that the provided new experiments and the additional explanation have convinced you of the merits of this paper. If there are additional questions, please feel free to let us know.
> >
> > Meanwhile, we would like to thank the reviewer again for the very helpful comments. By taking them into account, it would indeed make our paper clearer and stronger.
> >
> > Thank you for your time and effort!
> >
> > Best regards, Authors

---

> > > ### Author Response · Authors · 2021-11-30
> > > **A kind reminder**
> > >
> > > Dear reviewer bra9
> > >
> > > The interactive discussion phase will end in few hours, and we cannot have discussions with you anymore after the deadline. We wish that our response has addressed your concerns, and turns your assessment to a more positive side. Please let us know if there are any other things that we need to clarify.
> > >
> > > We thank you so much for your helpful and insightful suggestion.
> > >
> > > Best, Authors

---

> > > > ### Comment · Reviewer_bra9 · 2021-12-01
> > > > **Thanks for the comments**
> > > >
> > > > Please accept my apologies for the delay in responding.  The extra experiments and explanations are really appreciated. After reviewing the changes to this manuscript, I will opt to keep my score the same.

---

> > > > > ### Author Response · Authors · 2021-12-02
> > > > > **Thank you!**
> > > > >
> > > > > Thank you very much for your response. We are glad to learn that you have been satisfied with the additional experiments and discussion.

---

### Official Review · Reviewer_VThV · 2021-11-04

**Correctness:** 3
**Technical Novelty And Significance:** 2
**Empirical Novelty And Significance:** 2
**Recommendation:** 6
**Confidence:** 4

**Main Review:**

The paper is fairly organized, overall supported by theoretical foundations and setups the right experiments for evaluating unsupervised conditional generation GANs.

However, I find some of the paper's claims are neither trivial (or well-accepted IMO) nor are they supported/backed in the paper:

For example, the abstract says "However, existing unsupervised conditional GANs cannot cluster attributes of these data in their latent spaces properly because they assume uniform distributions of the attributes." This claim needs to be supported by some reasoning/intuition and evidence/references as to why this is true. I can argue that this is not necessarily true, for example, [Self-Cond-Gan](https://openaccess.thecvf.com/content_CVPR_2020/papers/Liu_Diverse_Image_Generation_via_Self-Conditioned_GANs_CVPR_2020_paper.pdf), shows that clustering is possible using D's features in an unsupervised manner.

Some of the novelties and tricks described in the paper (e.g. the reparametrization trick, or derivation and usage of the Theorem 1,2 are fairly trivial. However, the application of these ideas to Unsup GANs is a novel contribution of the paper.

The experiments are well designed, however, some of the right baselines have not been compared against.
The two models:

1- [Self-Cond-Gan](https://openaccess.thecvf.com/content_CVPR_2020/papers/Liu_Diverse_Image_Generation_via_Self-Conditioned_GANs_CVPR_2020_paper.pdf) which is an unsupervised conditional GAN, based on the DCGAN architecture, and
2- [PGM GAN](https://openaccess.thecvf.com/content/CVPR2021/papers/Armandpour_Partition-Guided_GANs_CVPR_2021_paper.pdf) which is also a GAN for unsupervised conditional image generation and also uses contrastive clustering

Are both missing in the comparisons, both in the literature review and empirical comparisons.

Since these two works are not discussed, based on my understanding and comparing the empirical results, It seems that SLOGAN does not achieve a comparable clustering performance to either (in terms of NMI over CIFAR-10) nor in terms of over all generation quality (FID). It would also be interesting to see how the models compare in terms of the newly introduced ICFID.



**Summary Of The Paper:**

The paper is concerned with unsupervised image generation using GANs. They devise a solution (SLOGAN) for addressing the problems current models have, especially when dealing with datasets with unbalanced features/attributes. In addition, to learn attributes from data and improve the conditional generation, it suggests using a new contrastive loss (U2C).



**Summary Of The Review:**

In summary, the paper needs some adjustments before it is ready for publication. The related works need to be updated with the relevant works (Self-Cond-GAN and PGMGAN mentioned above). The experiments and empirical results need to be compared with these relevant works or discussed why the comparison might not be fair.

EDIT: Given the proposed method is moderately novel and the newly presented comparisons after the discussions with authors, I increased my rating from 5 to 6.

---

> ### Author Response · Authors · 2021-11-19
> **Response to Reviewer VThV**
>
> We thank you for the insightful comments and suggestions. We have addressed each of your questions below.
>
> **Q1.** For example, the abstract says "However, existing unsupervised conditional GANs cannot cluster attributes of these data in their latent spaces properly because they assume uniform distributions of the attributes." This claim needs to be supported by some reasoning/intuition and evidence/references as to why this is true. I can argue that this is not necessarily true, for example, Self-Cond-Gan, shows that clustering is possible using D's features in an unsupervised manner.
>
> **Response:** Thank you for the insightful comment! Unsupervised conditional GANs such as InfoGAN, ClusterGAN, and CD-GAN assume uniform distributions of the attributes when the imbalance ratio is not given. As you pointed out, Self-conditioned GAN (and PGMGAN) does not assume uniform distributions. In the revision, we have added references in the introduction and have revised this claim more weakly considering Self-conditioned GAN and PGMGAN.
>
> **Q2.** The experiments are well designed, however, some of the right baselines have not been compared against. The two models: 1- Self-Cond-Gan which is an unsupervised conditional GAN, based on the DCGAN architecture, and 2- PGM GAN which is also a GAN for unsupervised conditional image generation and also uses contrastive clustering. Are both missing in the comparisons, both in the literature review and empirical comparisons.
>
> **Response:** We appreciate your suggestion. Self-conditioned GAN performs unsupervised conditional generation by using clustering of discriminator features as labels. PGMGAN applies a contrastive clustering method named SCAN [1] to perform unsupervised conditional generation. The pretrained space partitioner and a penalization loss function are used to encourage the generator to generate images with salient attributes.
>
> Following your suggestion, we conducted new comparative experiments with Self-conditioned GAN and PGMGAN and the results on balanced and imbalanced cifar datasets are summarized in the following tables:
>
> <Table S1.> Performance comparison on the CIFAR-2 dataset.
>
> |Method|NMI$\uparrow$|FID$\downarrow$|ICFID$\downarrow$|
> |-----------|:-----------:|:-----------:|:-----------:|
> |Self-conditioned GAN|0.00$\pm$0.00|39.44$\pm$1.72|71.54$\pm$5.41|
> |PGMGAN|0.67$\pm$0.00|29.49$\pm$0.51|**35.67$\pm$0.61**|
> |**SLOGAN**|**0.78$\pm$0.03**|**28.99$\pm$0.36**|35.68$\pm$0.51|
>
> <Table S2.> Performance comparison on the CIFAR-10 dataset.
>
> |Method|NMI$\uparrow$|FID$\downarrow$|ICFID$\downarrow$|
> |-----------|:-----------:|:-----------:|:-----------:|
> |Self-conditioned GAN|0.01$\pm$0.00|199.28$\pm$57.16|262.54$\pm$59.29|
> |PGMGAN|0.29$\pm$0.02|31.50$\pm$0.73|81.25$\pm$11.55|
> |**SLOGAN**|**0.34$\pm$0.01**|**20.61$\pm$0.40**|**71.23$\pm$6.76**|
>
> As shown in the results on the datasets with balanced attributes, SLOGAN outperforms Self-conditioned GAN and shows comparable performances with PGMGAN. Self-conditioned GAN has no loss to facilitate clustering of discriminator features, and the performance seems to be largely influenced by the architecture of the discriminator.
>
> <Table S3.> Performance comparison on the CIFAR-2 (7:3) dataset.
>
> |Method|NMI$\uparrow$|FID$\downarrow$|ICFID$\downarrow$|
> |-----------|:-----------:|:-----------:|:-----------:|
> |Self-conditioned GAN|0.00$\pm$0.00|45.28$\pm$1.81|88.58$\pm$4.57|
> |PGMGAN|0.42$\pm$0.03|29.76$\pm$1.65|57.06$\pm$3.31|
> |**SLOGAN**|**0.69$\pm$0.02**|**29.09$\pm$0.73**|**45.83$\pm$3.03**|
>
> <Table S4.> Performance comparison on the CIFAR-2 (9:1) dataset.
>
> |Method|NMI$\uparrow$|FID$\downarrow$|ICFID$\downarrow$|
> |-----------|:-----------:|:-----------:|:-----------:|
> |Self-conditioned GAN|0.00$\pm$0.00|50.45$\pm$1.56|123.35$\pm$6.56|
> |PGMGAN|0.16$\pm$0.03|30.23$\pm$1.31|101.68$\pm$3.87|
> |**SLOGAN**|**0.38$\pm$0.01**|**29.47$\pm$1.53**|**86.75$\pm$1.87**|
>
> As shown in the results on the datasets with imbalanced attributes, SLOGAN significantly outperforms the other methods. To avoid degenerated clusters where one partition contains most of the data, PGMGAN uses an entropy term that regularizes the average cluster probability vector to a uniform distribution. However, when learning PGMGAN on datasets with imbalanced attributes, it is difficult to adjust the coefficient of the entropy term because the performance seems to be sensitive to the strength of the entropy term in practice. We have added results on more datasets in Tables 1, 2, 6, and 7 in the revision. We also have cited these works to the background and discussed the methods in Appendix B.2 in the revision.
>
> We wish that our response has addressed your concerns. If you have any questions, please feel free to let us know during the rebuttal window. Thank you very much!
>
> [1] SCAN: Learning to Classify Images Without Labels, ECCV 2020.

---

> > ### Comment · Reviewer_VThV · 2021-11-22
> > **What is the source of the numbers?**
> >
> > I would like to thank the authors for their detailed response.
> >
> > Looking at the selfcondgan and pgmgan papers, the reported results do not match the ones in the provided tables (both papers report on CIFAR-10), I am interested to know what is the difference?
> >
> > The authors brought up an interesting point, how sensitive is SLOGAN to the coefficient of U2C loss?

---

> > > ### Author Response · Authors · 2021-11-22
> > > **Response**
> > >
> > > Thank you very much for your additional comments!
> > >
> > > **Q1.** Looking at the selfcondgan and pgmgan papers, the reported results do not match the ones in the provided tables (both papers report on CIFAR-10), I am interested to know what is the difference?
> > >
> > > **Response:** Our experimental results and the reported results are different in two aspects.
> > >
> > > First, for a fair comparison, we used the same network architecture and hyperparameters (e.g., learning rates) across all methods. For PGMGAN, we used the encoder's backbone architecture described in Appendix E.5 as the backbone architecture of the space partitioner and added pretext heads, a partitioner head, and spectral normalization.
> > >
> > > Second, the objective of our work is conditional generation on salient attributes of data. Therefore, we set the number of clusters equal to the number of classes in the dataset to measure ICFID which is the performance metric for unsupervised conditional generation. The reported results in Table 3 of Self-conditioned GAN paper also show that Self-conditioned GAN with k=10 shows worse FID than vanilla GAN on the CIFAR-10 dataset.
> > >
> > >
> > > **Q2.** The authors brought up an interesting point, how sensitive is SLOGAN to the coefficient of U2C loss?
> > >
> > > **Response:** We appreciate your suggestion. We conducted ablation studies to compare how sensitive the performance is according to the coefficient of the entropy term of PGMGAN and the coefficient of U2C loss of SLOGAN. The results on the CIFAR-2 (7:3) dataset are summarized in the following tables:
> > >
> > > <Table S5.> Performance of PGMGAN on CIFAR-2 (7:3) with different coefficients of the entropy term
> > >
> > > |Coefficient| 2| 1| 0.5|
> > > |-----------|:-----------:|:-----------:|:-----------:|
> > > |$\pi_{y=0}$ (ground truth=0.7)|0.52|0.58|1.00|
> > > |NMI $\uparrow$|0.36|**0.42**|0.00|
> > >
> > > <Table S6.> Performance of SLOGAN on CIFAR-2 (7:3) with different coefficients of U2C loss
> > >
> > > |Coefficient| 2| 1| 0.5|
> > > |-----------|:-----------:|:-----------:|:-----------:|
> > > |$\pi_{y=0}$ (ground truth=0.7)|0.71|0.69|0.68|
> > > |NMI $\uparrow$|0.63|**0.69**|0.62|
> > >
> > > From the results, lowering the coefficient of the entropy term of PGMGAN from 1 to 0.5 results in degenerated clusters. Therefore, PGMGAN seems to be sensitive to the coefficient of the entropy term when trained on imbalanced datasets. On the other hand, we confirmed that SLOGAN shows robustly high performance on imbalanced datasets even when the coefficient of U2C loss is changed.

---

> > > > ### Author Response · Authors · 2021-11-27
> > > > **Looking forward to your follow-up response!**
> > > >
> > > > Dear Reviewer VThV
> > > >
> > > > Thank you again for the insightful comments and suggestions! Since the deadline of discussion is approaching, we sincerely look forward to your follow-up response. We are happy to provide any additional clarification that you may need.
> > > >
> > > > In our previous response, we conducted additional experiments to compare our method with the relevant works (Self-conditioned GAN and PGMGAN). For your convenience, we provide a summary below:
> > > >
> > > > - Conducted the new experiments to show that SLOGAN outperforms Self-conditioned GAN and PGMGAN.
> > > > - Provided additional discussion on Self-conditioned GAN and PGMGAN.
> > > > - Conducted the new experiments to show that SLOGAN is more robust to the changes in the coefficient of the loss than PGMGAN.
> > > >
> > > > We hope that the provided new experiments and the additional discussion on Self-conditioned GAN and PGMGAN have convinced you of the merits of this paper. Please do not hesitate to contact us if there are additional questions.
> > > >
> > > > Meanwhile, we would like to thank the reviewer again for the very helpful comments. By taking them into account, it would indeed make our paper clearer and stronger.
> > > >
> > > > Thank you for your time and effort!
> > > >
> > > > Best regards, Authors

---

> > > > > ### Comment · Reviewer_VThV · 2021-12-02
> > > > > **Thanks for the comments**
> > > > >
> > > > >  Given the proposed method is moderately novel and the newly presented comparisons after the discussions with authors, I increased my rating from 5 to 6.

---

> > > > > > ### Author Response · Authors · 2021-12-02
> > > > > > **Thank you!**
> > > > > >
> > > > > > We appreciate your response. We are sincerely glad to learn that you have been satisfied with the additional comparisons and discussion.

---

### Author Response · Authors · 2021-11-19
**General Response**

We thank all the reviewers very much for their valuable comments and constructive suggestions to strengthen our work. Following reviewers’ suggestions, we have added literature reviews and more supporting experiments in the revision. Here, we would like to highlight major changes in the revision:

### Literature reviews
- We cited and discussed more papers including NEMGAN [1], Self-conditioned GAN [2], PGMGAN [3] suggested by Reviewers VThV and 5GSd: Appendix B.2

### Experiments
- Comparison with Self-conditioned GAN and PGMGAN [VThV, 5GSd]: Tables 1, 2, 6, and 7, Appendix B.2
- Qualitative comparison with the most recent methods [bra9]: Figure 20, Appendix A.6
- More quantitative results of SLOGAN without U2C loss [5GSd]: Table 9
- Synthetic datasets with various covariances [9K6Q]: Figure 11
- Highly imbalanced multi-class MNIST dataset [bra9, 5GSd]: Figure 21, Appendix A.6
- AFHQ dataset with various imbalance ratios [bra9, 9K6Q]: Figure 3 (a), Figure 22, Appendix A.6

[1] Effect of The Latent Structure on Clustering with GANs, IEEE Signal Processing Letters 2020.\
[2] Diverse image generation via self-conditioned gans, CVPR 2020.\
[3] Partition-Guided GANs, CVPR 2021.

---

### Decision · Program_Chairs · 2022-01-20

**Decision:**

Accept (Poster)

**Comment:**

This paper proposes an unsupervised learning method for GANs, called SLOGAN, which allows conditional generation of samples, by utilizing clustering structures of training data in a latent space. The main significance of the proposal over existing unconditional conditional GANs is that it is capable of dealing with training data with imbalance in the latent space. The proposal consists of the use of implicit reparameterization based on the generalized Stein lemma, which makes learning of the mixing coefficient parameters possible, as well as introduction of the U2C loss.

The initial review score distribution is such that two of them are just above the acceptance threshold, and two others are just below it. Upon reading the review comments and the author responses, as well as the paper itself, I think that the evaluations of the reviewers are more or less coherent with each other:

1. The proposed method is moderately, if not significantly, novel: The differences from DeLiGAN are the use of implicit reparameterization based on the generalized Stein lemma, learning of the mixing coefficient parameters, and introduction of the U2C loss.
2. The experimental results, while demonstrating effectiveness of the proposed method to some extent, were not convincing enough.

As for the item 2, the authors have provided results of additional experiments in their responses, as suggested by the reviewers, and two reviewers have revised their scores upward accordingly.

Yet another point I would like to mention is that in some numerical results summarized in Tables 1 and 2, as well as in several other places, one can notice somewhat large errors, so that one might be able to question the statistical significance of the claimed best-performing methods, shown in bold. (If my guess would be correct, the authors regarded the *best in the mean* as the best, ignoring the standard error, and did not perform any statistical testing to confirm the significance.) I would therefore appreciate additional assessment of significance of the numerical results via proper statistical testing.

Because of the above, I would like to recommend acceptance of this paper.